# Reduced discrimination between signals of danger and safety but not overgeneralization is linked to exposure to childhood adversity in healthy adults

Maren Klingelhöfer-Jens[1]*, Katharina Hutterer[2], Miriam A Schiele[3], Elisabeth J Leehr[4], Dirk Schümann[1], Karoline Rosenkranz[1], Joscha Böhnlein[4], Jonathan Repple[4,5], Jürgen Deckert[2], Katharina Domschke[3], Udo Dannlowski[4], Ulrike Lueken[6,7], Andreas Reif[5,8], Marcel Romanos[9], Peter Zwanzger[10,11], Paul Pauli[12], Matthias Gamer[12], Tina B Lonsdorf[13]

[1]Institute for Systems Neuroscience, University Medical Center Hamburg-Eppendorf, Hamburg, Germany; [2]Department of Psychiatry, Psychosomatics and Psychotherapy, Center of Mental Health, University Hospital of Würzburg, University of Würzburg, Würzburg, Germany; [3]Department of Psychiatry and Psychotherapy, Medical Center - University of Freiburg, Faculty of Medicine, University of Freiburg, Freiburg, Germany; [4]Institute for Translational Psychiatry, University of Münster, Münster, Germany; [5]Department of Psychiatry, Psychosomatic Medicine and Psychotherapy, University Hospital Frankfurt – Goethe University, Frankfurt am Main, Germany; [6]Department of Psychology, Humboldt-Universität zu Berlin, Berlin, Germany; [7]German Center of Mental Health (DZPG), partner site Berlin-Potsdam, Berlin, Germany; [8]Fraunhofer Institute for Translational Medicine and Pharmacology ITMP, Frankfurt am Main, Germany; [9]Department of Child and Adolescent Psychiatry, Psychosomatics and Psychotherapy, Center of Mental Health, University Hospital of Würzburg, University of Würzburg, Würzburg, Germany; [10]Kbo Inn Salzach Hospital Clinical Center for Psychiatry, Wasserburg am Inn, Germany; [11]Department of Psychiatry, Ludwig-Maximilian-University Munich, Munich, Germany; [12]Department of Psychology and Center of Mental Health, Julius-Maximilians-University of Würzburg, Würzburg, Germany; [13]Department of Psychology, Biological Psychology and Cognitive Neuroscience, University of Bielefeld, Bielefeld, Germany

*For correspondence: m.klingelhoefer-jens@uke.de

Competing interest: The authors declare that no competing interests exist.

## eLife Assessment

This **important** study addresses two questions: (i) how danger signaling is altered for people with childhood adversities, and (ii) how this differs across different operationalizations of adversity. The latter is of particularly broad interest to multiple fields, given that childhood adversity is operationalized very differently across the literature. The study provides **compelling** evidence using a large sample size and rigorous statistical methods. These data will be of interest to scientists and clinicians interested in early life adversity, statistical approaches for quantifying stress exposure, or aversive learning.

**Abstract** Childhood adversity is a strong predictor of developing psychopathological conditions. Multiple theories on the mechanisms underlying this association have been suggested which,

however, differ in the operationalization of 'exposure.' Altered (threat) learning mechanisms represent central mechanisms by which environmental inputs shape emotional and cognitive processes and ultimately behavior. 1402 healthy participants underwent a fear conditioning paradigm (acquisition training, generalization), while acquiring skin conductance responses (SCRs) and ratings (arousal, valence, and contingency). Childhood adversity was operationalized as (1) dichotomization, and following (2) the specificity model, (3) the cumulative risk model, and (4) the dimensional model. Individuals exposed to childhood adversity showed blunted physiological reactivity in SCRs, but not ratings, and reduced CS+/CS- discrimination during both phases, mainly driven by attenuated CS+ responding. The latter was evident across different operationalizations of 'exposure' following the different theories. None of the theories tested showed clear explanatory superiority. Notably, a remarkably different pattern of increased responding to the CS- is reported in the literature for anxiety patients, suggesting that individuals exposed to childhood adversity may represent a specific sub-sample. We highlight that theories linking childhood adversity to (vulnerability to) psychopathology need refinement.

## Introduction

Exposure to childhood adversity - particularly in early life - is a strong predictor for developing somatic, psychological, and psychopathological conditions (*Anda et al., 2006*; *Baldwin et al., 2024*; *Danese et al., 2007*; *Danese and Widom, 2023*; *Felitti, 2002*; *Gilbert et al., 2009*; *Green et al., 2010*; *Heim and Nemeroff, 2002*; *Hosseini-Kamkar et al., 2023*; *Klauke et al., 2010*; *McLaughlin et al., 2012*; *Moffitt et al., 2007*; *Samaey et al., 2024*; *Teicher et al., 2022*) and is hence associated with substantial individual suffering as well as societal costs (*Hughes et al., 2021*). Exposure to childhood adversity is rather common, with nearly two-thirds of individuals experiencing one or more traumatic events prior to their 18[th] birthday (*McLaughlin et al., 2013*). While not all trauma-exposed individuals develop psychopathological conditions, there is some evidence of a dose-response relationship (*Danese et al., 2009*; *Smith and Pollak, 2021*; *Young et al., 2019*). As this potential relationship is not yet fully clear, understanding the mechanisms by which childhood adversity becomes biologically embedded and contributes to the pathogenesis of stress-related somatic and mental disorders is central to the development of targeted intervention and prevention programmes. Learning is a core mechanism through which environmental inputs shape emotional and cognitive processes and ultimately behavior. Thus, learning mechanisms are key candidates potentially underlying the biological embedding of exposure to childhood adversity and their impact on development and risk for psychopathology (*McLaughlin and Sheridan, 2016*).

Fear conditioning is a prime translational paradigm for testing potentially altered (threat) learning mechanisms following exposure to childhood adversity under laboratory conditions. The fear conditioning paradigm typically consists of different experimental phases (*Lonsdorf et al., 2017*). During fear acquisition training, a neutral cue is paired with an aversive event such as an electrotactile stimulation or a loud aversive human scream (unconditioned stimulus, US). Through these pairings, an association between both stimuli is formed and the previously neutral cue becomes a conditioned stimulus (CS+) that elicits conditioned responses. In human differential conditioning experiments, typically a second neutral cue is never paired with the US and serves as a control or safety stimulus (i.e. CS-). During a subsequent fear extinction training phase, both the CS+ and the CS- are presented without the US which leads to a gradual waning of conditioned responding. A fear generalization phase includes additional stimuli (i.e. generalization stimuli; GSs) that are perceptually or conceptually similar to the CS+ and CS- (e.g. generated through merging perceptual properties of the CS+ and CS-) which allows for the investigation of the degree to which conditioned responding generalizes to similar cues.

Fear acquisition as well as extinction are considered as experimental models of the development and exposure-based treatment of anxiety- and stress-related disorders. Fear generalization is in principle adaptive in ensuring survival ('better safe than sorry'), but broad overgeneralization can become burdensome for patients. Accordingly, maintaining the ability to distinguish between signals of danger (i.e. CS+) and safety (i.e. CS-) under aversive circumstances is crucial, as it is assumed to be beneficial for healthy functioning (*Hölzel et al., 2016*) and predicts resilience to life stress (*Craske et al., 2012*), while reduced discrimination between the CS+ and the CS- has been linked to pathological anxiety

(*Duits et al., 2015*; *Lissek et al., 2005*): Meta-analyses suggest that patients suffering from anxiety- and stress-related disorders show enhanced responding to the safe CS- during fear acquisition (*Duits et al., 2015*). During extinction, patients exhibit stronger defensive responses to the CS+ and a trend toward increased discrimination between the CS+ and CS- compared to controls, which may indicate delayed and/or reduced extinction (*Duits et al., 2015*). Furthermore, meta-analytic evidence also suggests stronger generalization to cues similar to the CS+ in patients and more linear generalization gradients (*Cooper et al., 2022*; *Dymond et al., 2015*; *Fraunfelter et al., 2022*). Hence, aberrant fear acquisition, extinction, and generalization processes may provide clear and potentially modifiable targets for intervention and prevention programs for stress-related psychopathology (*McLaughlin and Sheridan, 2016*).

In sharp contrast to these threat learning patterns observed in patient samples, a recent review provided converging evidence that exposure to childhood adversity is linked to reduced CS discrimination, driven by blunted responding to the CS+ during experimental phases characterized through the presence of threat (i.e. acquisition training and generalization, *Ruge et al., 2024*). Of note, this pattern was observed in mixed samples (healthy, at risk, patients), in pediatric samples, and in adults exposed to childhood adversity as children. The latter suggests that recency of exposure or developmental timing may not play a major role, even though there is some evidence pointing towards accelerated pubertal and neural (connectivity) development in exposed children (*Machlin et al., 2019*; *Silvers et al., 2016*). There is, however, no evidence pointing towards differences in extinction learning or generalization gradients between individuals exposed and unexposed to childhood adversity (for a review, see *Ruge et al., 2024*).

*Ruge et al., 2024* also highlighted operationalization as a key challenge in the field hampering the interpretation of findings across studies and consequently cumulative knowledge generation. Operationalization of exposure to childhood adversity, and hence the translation of theoretical accounts of the role of childhood adversity into statistical tests, is an ongoing discussion in the field (*McLaughlin et al., 2021*; *Pollak and Smith, 2021*; *Smith and Pollak, 2021*). Historically, childhood adversity has been conceptualized rather broadly considering different adversity types lumped into a single category. This follows from the (implicit) assumption that any exposure to an adverse event will have similar and additive effects on the individual and its (neuro-biological) development (*Smith and Pollak, 2021*). Accordingly, childhood adversity has often been considered as a cumulative measure ('cumulative risk approach;' *McLaughlin et al., 2021*; *Smith and Pollak, 2021*). An alternative approach posits that different types of adverse events have a distinct impact on individuals and their (neuro-biological) development through distinct mechanisms ('specificity approach;' *McLaughlin et al., 2021*; *Sheridan and McLaughlin, 2014*; *Smith and Pollak, 2021*). Currently, distinguishing between threat and deprivation exposure represents the prevailing approach (*McLaughlin et al., 2019a*), which has been formalized in the (two-)dimensional model of adversity and psychopathology (DMAP; *Machlin et al., 2019*; *McLaughlin and Sheridan, 2016*; *McLaughlin et al., 2021*; *McLaughlin et al., 2014*; *Sheridan and McLaughlin, 2014*; *McLaughlin et al., 2016*; *Sheridan and McLaughlin, 2016*). To this end, exposure to threat-related childhood adversity has been suggested to be specifically linked to altered emotional and fear learning (*Sheridan and McLaughlin, 2014*).

Yet, there is converging evidence from different fields of research suggesting that the effects of exposure to childhood adversity are cumulative, non-specific, and rather unlikely to be tied to specific types of adverse events (*Danese et al., 2009*; *Smith and Pollak, 2021*; *Young et al., 2019*) - with few exceptions (*Colich et al., 2020*; *McLaughlin et al., 2019b*). This is also supported by the recent review of *Ruge et al., 2024*. However, the different theoretical accounts have not yet been directly compared in a single fear conditioning study.

Here, we aim to fill this gap in an extraordinarily large sample of healthy adults (N=1402) recruited in the context of a multi-centric study conducted at the Universities of Münster, Würzburg, and Hamburg, Germany. Participants underwent a differential fear conditioning paradigm including a fear acquisition and generalization phase using female faces as CSs and GSs and a female scream as US, while we measured skin conductance responses (SCRs) and different ratings types (i.e. arousal, valence, and US contingency). For SCRs and fear ratings, we calculated three different outcomes: CS discrimination (i.e. the difference between CS+ and CS- responses), the linear deviation score (LDS) as an index of the linearity of the generalization gradient (*Kaczkurkin et al., 2017*), and the general reactivity which was defined as the reactivity across all phases (for more details, see Materials and

methods section). We also performed manipulation checks to verify whether the experimental manipulations had the intended effect.

We operationalized childhood adversity exposure through different approaches: Our main analyses employ the approach adopted by most publications in the field (see *Ruge et al., 2024* for a review) - dichotomization of the sample into exposed vs. unexposed individuals based on published cut-offs for the Childhood Trauma Questionnaire (CTQ; *Bernstein et al., 2003*; *Wingenfeld et al., 2010*). Individuals were classified as exposed to childhood adversity if at least one CTQ subscale met the published cut-offs (*Bernstein and Fink, 1998*; *Häuser et al., 2011*) for at least moderate exposure (i.e. emotional abuse ≥ 13, physical abuse ≥ 10, sexual abuse ≥ 8, emotional neglect ≥ 15, physical neglect ≥ 10).

In addition, we provide exploratory analyses that attempt to translate dominant (verbal) theoretical accounts (*McLaughlin et al., 2021*; *Pollak and Smith, 2021*) on the impact of exposure to childhood adversity into statistical tests. At the same time, we acknowledge that such a translation is not unambiguous and these exploratory analyses should be considered as showcasing a set of plausible solutions. With this, we aim to facilitate comparability, replicability, and cumulative knowledge generation in the field as well as providing a solid base for hypothesis generation (*Ruge et al., 2024*) and refinement of theoretical accounts. More precisely, we attempted to exploratively translate (a) the cumulative risk approach, which is based on the assumed key role of cumulative childhood adversity exposure, (b) the specificity model, which considers specific types of exposure (in the present study: abuse and neglect), and (c) the dimensional model, which also considers specific exposure types but controls for the effects of one another, into statistical tests applied to our dataset. Furthermore, we compiled challenges that arise in the practical implementation of these verbal theories into statistical models (for more details, see *Table 1*).

Based on the recently reviewed literature (*Ruge et al., 2024*), we expected less discrimination between signals of danger (CS+) and safety (CS-) in exposed individuals as compared to those unexposed to childhood adversity - primarily due to reduced CS+ responses - during both the fear acquisition and the generalization phase. Based on the literature (*Ruge et al., 2024*), we did not expect group differences in generalization gradients.

## Results

Exposed and unexposed participants were equally distributed across data recording sites ($\chi^2(3)$=3.72, p=0.293).

### Main effect of task

In brief, and as reported previously (*Herzog et al., 2021*; *Schiele et al., 2016a*), the fear acquisition was successful in SCRs as well as ratings in the full sample (all *p*'s<0.001; see Appendix 1 for details). During fear generalization, the expected generalization gradient was observed with a gradual increase in SCRs and ratings with increasing similarity to the CS+ (all *p*'s<0.01 except for the comparisons of SCRs to CS- vs. GS4 as well as GS1 vs. GS2 which were non-significant; see Appendix 1).

### Association between different outcomes and exposure to childhood adversity

During both the acquisition training and generalization phase, CS discrimination in SCRs was significantly lower in individuals exposed to childhood adversity as compared to unexposed individuals (see *Table 2* and *Figure 1*; for trial-by-trial responses, see *Appendix 1—figure 4*). Post hoc analyses (i.e. ANOVAs) revealed that childhood adversity exposure significantly interacted with stimulus type (acquisition training: $F(1, 1400)$=5.42, p=0.020, $\eta_p^2$<0.01; generalization test: $F(1, 1051)$=5.37, p=0.021): SCRs to the CS + during both acquisition training and the generalization phase were significantly lower in exposed as compared to unexposed individuals (acquisition training: $t(1400)$=2.54, p=0.011, *d*=0.14; generalization test: $t$=(194.1)=3.51, p=0.001, *explanatory measure of effect size*=0.179; see *Figure 2*) but not for the CS- (acquisition training: $t(1400)$=0.75, p=0.452, *d*=0.04; generalization test: $t(178.9)$=1.63, p=0.104, *explanatory measure of effect size*=0.09). For ratings, no significant effects of exposure to childhood adversity were observed in CS discrimination (see *Table 2*).

**Table 1.** Operationalization of childhood adversity in different theoretical approaches and challenges of their statistical translation.

| Approach name and reference | Operationalization of childhood adversity | Challenges in translating theory into a statistical model |
|---|---|---|
| **Main analyses** | | |
| Moderate exposure based on CTQ (exposed vs. unexposed) | **Short description**: At least one subscale met the published cut-off for at least moderate exposure (**Bernstein and Fink, 1998**; **Häuser et al., 2011**). The moderate cut-off was chosen, as it was recently identified as the most commonly used in the literature (for a review see **Ruge et al., 2024**)<br>**Procedure**: Dichotomization of the sample into exposed vs. unexposed individuals based on published cut-offs: emotional abuse ≥ 13, physical abuse ≥ 10, sexual abuse ≥ 8, emotional neglect ≥ 15, physical neglect ≥ 10. Such cut-off of moderate exposure was employed in previous work by our team (**Koppold et al., 2023**) and in the literature (**Ruge et al., 2024**)<br>**Statistical test**: See Materials and methods: Statistical analyses | • Not based on an existing theory but on what is commonly used in the literature (**Ruge et al., 2024**)<br>• Different cut-offs published (for a discussion, see **Ruge et al., 2024**)<br>• (Statistical) Challenges linked to dichotomization of an inherently continuous variable |
| **Exploratory analyses** | | |
| Cumulative risk model (**Evans et al., 2013**; **McEwen, 2003**) | **Short description**: Based on the assumed key role of cumulative exposure (exposure intensity and frequency)<br>**Procedure (a)**: Classification into four severity groups (no, low, moderate, severe exposure) based on cut-offs published by **Bernstein and Fink, 1998**<br>**Statistical test (a)**: Comparison of conditioned responding of the four severity groups by using one-way ANOVAs<br>**Procedure (b)**: Number of subscales exceeding an at least moderate cut-off based on **Bernstein and Fink, 1998** and **Häuser et al., 2011**<br>**Statistical test (b)**: Number of sub-scales exceeding an at least moderate cut-off as a predictor and conditioned responding as the criterion in simple linear regression models | • Problem with CTQ sum score: it assigns the same 'value' to all CM types (see also 'General operationalizational challenges' below)<br>• Number of subscales exceeding cut-off: calculate ANOVA or regression?<br>• Cumulative risk scores are based on the implicit assumption that different types of adverse events affect the same mechanisms and are of equal impact |
| Specificity model (**McMahon et al., 2003**; **Pollak et al., 2000**; **Pollak and Tolley-Schell, 2004**) | **Short description**: Consideration of specific exposure types (abuse vs. neglect)<br>**Procedure**: Summing up the CTQ subscales of emotional abuse, physical abuse, and sexual abuse yielding a composite score for exposure to 'abuse' and summing up the subscales of emotional neglect and physical neglect to yield a composite score for 'neglect' (or threat vs. deprivation as done by **Sheridan et al., 2017**)<br>**Statistical test**: The abuse and neglect composite scores are tested for associations with conditioned responding in separate regression models.<br>In our sample, n=52 and n=96 individuals were exposed to abuse only and neglect only, respectively, while n=55 reported to have experienced both abuse and neglect. We included all participants in all analyses as done previously (**Sheridan et al., 2017**) | • What qualifies as a specific exposure type? (i.e. subscales or composite scales for neglect vs. abuse?)<br>• Which exposure subcategories are 'too specific' or 'too broad'? (A heterogeneous category may obscure potentially relevant discrete associations)<br>• Include only participants who experienced only one specific type but not any other types despite this being rather artificial due to high co-occurrences of different exposure types and requiring extremely large samples? Which cut-off should be used then to define exposure? We decided to include all participants in the analyses as done in previous studies (**Sheridan et al., 2017**)<br>• Lack of specificity of exposure subtypes (e.g. sexual abuse also has an emotional component) |

*Table 1 continued on next page*

*Table 1 continued*

| Approach name and reference | Operationalization of childhood adversity | Challenges in translating theory into a statistical model |
|---|---|---|
| **Dimensional model** (**McLaughlin et al., 2016**; **McLaughlin et al., 2021**) | **Short description:** Consideration of specific exposure types (i.e. abuse and neglect) that are assumed to co-occur and be controlled for the effect of one another (as opposed to the specificity model) **Procedure:** See specificity model **Statistical test:** Abuse and neglect scores are tested for associations with conditioned responding in a single linear regression model in which the influence of the other exposure type is controlled for | • Ongoing debate on the multicollinearity of multiple types of childhood adversity in one model (**McLaughlin et al., 2021; Pollak and Smith, 2021**) |
| **General operationalizational challenges** | • Non-comparability of dimensional and categorical approaches: CTQ sum score assumes an equal contribution of all items which contradicts different thresholds for being considered as exposed for different subscales (e.g. lower cut-off for sexual abuse as compared to emotional neglect)<br>• Associations in a full sample may differ from associations in the group of exposed individuals only which is a challenge for the interpretation of data<br>• Multiple cut-offs published (**Bernstein et al., 1997; Bernstein and Fink, 1998**)<br>• Specific challenges relating to abuse and neglect: They<br> • often co-occur<br> • are not the only relevant dimensions (e.g. unpredictability, loss)<br> • are not strongly supported as distinct dimensions in the literature (**Carozza et al., 2022; Smith and Pollak, 2021**)<br>• Heterogeneity in the assessment of childhood adversity across studies - both with respect to the operationalization of adversity across studies - both with respect to the operationalization of adversity (i.e. definition)<br>• Different response formats (yes/no vs. specification of duration and frequency) and the number of trauma types/events included in assessment tools impact on prevalence rates and potentially also associations between the number of adverse experiences and symptom severity (e.g. **Contractor et al., 2018**)<br>• Distinction between stressful events and trauma is often unclear (**Richter-Levin and Sandi, 2021**) | |

**Table 2.** Results of t-tests comparing conditioned stimulus (CS) discrimination, the linear deviation score (i.e. strength of generalization), and general reactivity between exposed and unexposed participants.

| Outcome | Phase | Measure | t | df | p | Cohen's d | LL (95% CI) | UL (95% CI) |
|---|---|---|---|---|---|---|---|---|
| CS discrimination | ACQ | SCR | **2.33** | 1,400 | **0.020** | –0.18 | –0.33 | –0.03 |
| | | Arousal ratings | –1.52 | 1,400 | 0.128 | 0.12 | –0.03 | 0.26 |
| | | Valence ratings | 0.20 | 1,400 | 0.845 | –0.01 | –0.16 | 0.13 |
| | | Contigency ratings | 0.70 | 1,400 | 0.484 | –0.05 | –0.20 | 0.10 |
| | GEN | SCR | **2.34** | 1,400 | **0.020** | –0.18 | –0.33 | –0.03 |
| | | Arousal ratings | –0.28 | 1,400 | 0.777 | 0.02 | –0.13 | 0.17 |
| | | Valence ratings | 0.06 | 1,400 | 0.953 | 0.00 | –0.15 | 0.14 |
| | | Contigency ratings | 0.58 | 1,400 | 0.560 | –0.04 | –0.19 | 0.10 |
| LDS | GEN | SCR | 1.41 | 295 | 0.158 | –0.10 | –0.25 | 0.05 |
| | | Arousal ratings | –0.62 | 1,400 | 0.538 | 0.05 | –0.10 | 0.20 |
| | | Valence ratings | 0.30 | 1,400 | 0.765 | –0.02 | –0.17 | 0.13 |
| | | Contigency ratings | –0.95 | 1,400 | 0.344 | 0.07 | –0.08 | 0.22 |
| General reactivity | ALL | SCR | **2.06** | 1,400 | **0.040** | –0.16 | –0.31 | –0.01 |
| | | Arousal ratings | –0.10 | 1,400 | 0.920 | 0.01 | –0.14 | 0.16 |
| | | Valence ratings | 0.83 | 1,400 | 0.408 | –0.06 | –0.21 | 0.09 |
| | | Contigency ratings | –0.97 | 250 | 0.334 | 0.07 | –0.09 | 0.24 |

Note. ACQ = acquisition training, GEN = generalization phase, LDS = linear deviation score. Bold numbers indicate significant results (p<0.05).

No significant effect of exposure to childhood adversity in either SCRs or ratings was observed for generalization gradients (see *Table 2* and *Figure 3*). It is, however, also evident from the generalization gradients that both groups differ specifically in reactivity to the CS+ (see above and *Figure 3*).

In addition, general physiological reactivity in SCRs (i.e. raw amplitudes) was significantly lower in participants exposed to childhood adversity compared to unexposed participants (see *Table 2* and *Figure 4*) while there were no differences between both groups in general rating response levels (see *Table 2*).

At the request of a reviewer, we repeated our main analyses by using linear mixed models including age, sex, school degree (i.e. to approximate socioeconomic status, SES), and exposure to childhood adversity as fixed effects as well as site as random effect. These analyses yielded comparable results demonstrating a significant effect of childhood adversity on CS discrimination during acquisition training and the generalization phase as well as on general reactivity, but not on the generalization gradients in SCRs (see *Appendix 1—table 2A*). Consistent with the results of the main analyses reported in our manuscript, we did not observe any significant effects of childhood adversity on the different types of ratings when using mixed models (see *Appendix 1—table 2B–D*). Some of the mixed model analyses showed significantly lower CS discrimination during acquisition training and generalization, and lower general reactivity in males compared to females (see *Appendix 1—table 2* for details).

## Exploratory analyses

The cumulative risk model operationalized through the different cut-offs for no, low, moderate, and severe exposure (*Bernstein and Fink, 1998*) did not yield any significant results for any outcome measure and experimental phase (see *Appendix 1—table 3*). However, on a descriptive level (see *Figure 5*), it seems that indeed exposure to an at least moderate cut-off level may induce behavioral and physiological changes (see main analysis, *Bernstein and Fink, 1998*). This might suggest that the cut-off for exposure commonly applied in the literature (see *Ruge et al., 2024*) may indeed represent a reasonable approach.

Cumulative risk operationalized as the number of CTQ subscales exceeding the moderate cut-off (*Bernstein and Fink, 1998*), however, revealed that a higher number of subscales exceeding the cut-off predicted lower CS discrimination in SCRs ($F_{(1, 1400)}$=6.86, p=0.009, $R^2$=0.005) and contingency ratings ($F_{(1, 1400)}$=4.08, p=0.044, $R^2$=0.003) during acquisition training (see *Appendix 1—table 4* and *Figure 6* for an exemplary illustration of SCRs during acquisition training). This was driven by significantly lower SCRs to the CS+ ($F_{(1, 1400)}$=5.42, p=0.02, $R^2$=0.004) while for contingency ratings no significant post hoc tests were identified (all p>0.05). For an illustration of how the different adversity

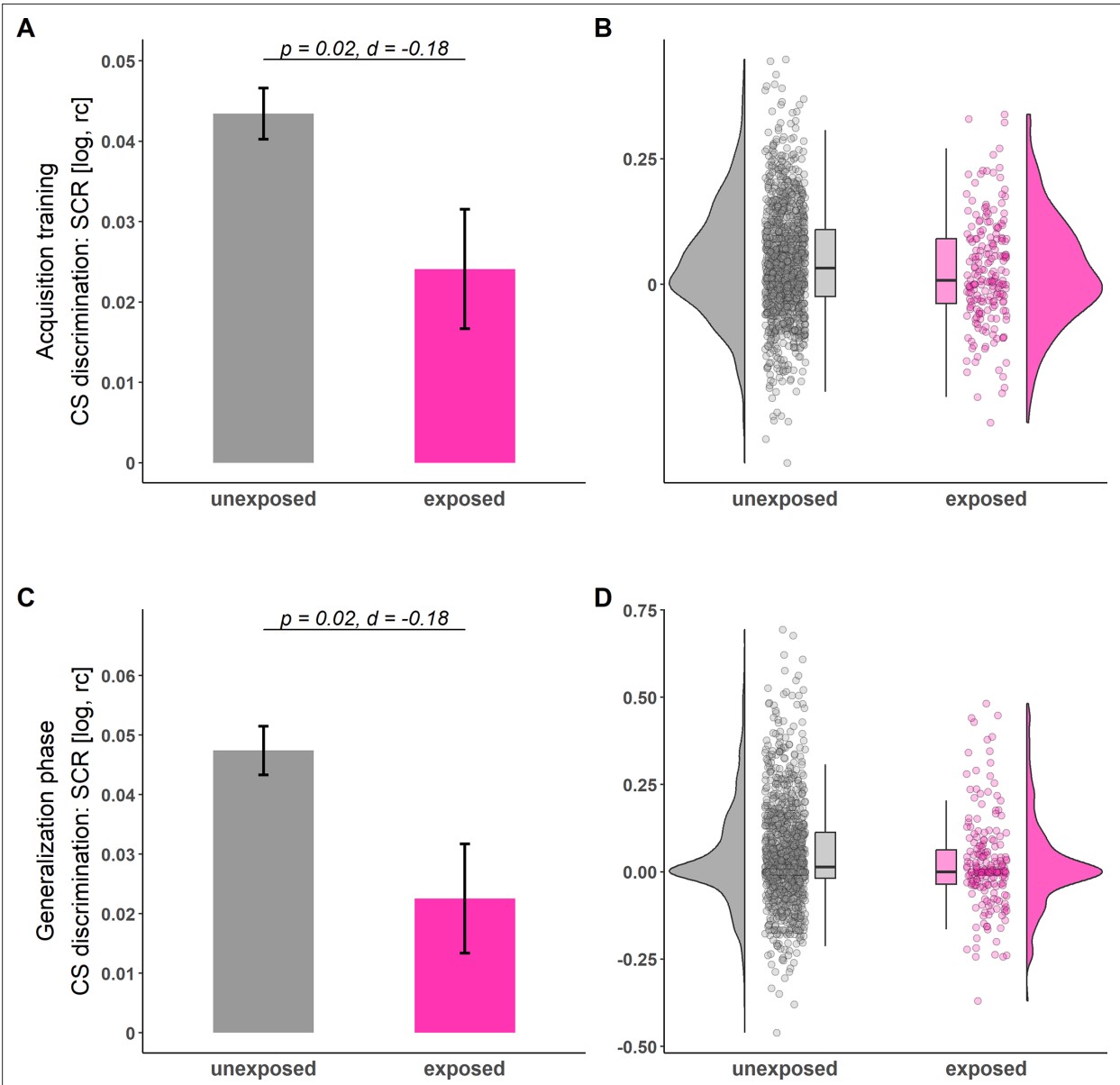

**Figure 1.** Illustration of conditioned stimulus (CS) discrimination in skin conductance responses (SCRs) during acquisition training (**A–B**) and generalization phase (**C–D**) for individuals unexposed (gray) and exposed (pink) to childhood adversity. Barplots (**A and C**) with error bars represent means and standard errors of the means (SEMs) including $n_{unexposed}$ = 1199 and $n_{exposed}$ = 203, respectively. The statistical parameters presented in **A** and **C** are derived from two-tailed independent-samples t-tests. The a priori significance level was set to α = 0.05. Distributions of the data are illustrated in the raincloud plots (**B and D**). Points next to the densities represent the CS discrimination of each participant averaged across phases. Boxes of boxplots represent the interquartile range (IQR) crossed by the median as a bold line, ends of whiskers represent the minimum/maximum value in the data within the range of 25th/75th percentiles ± 1.5 IQR. For trial-by-trial SCRs across all phases, see *Appendix 1—figure 4*. log = log-transformed, rc = range-corrected.

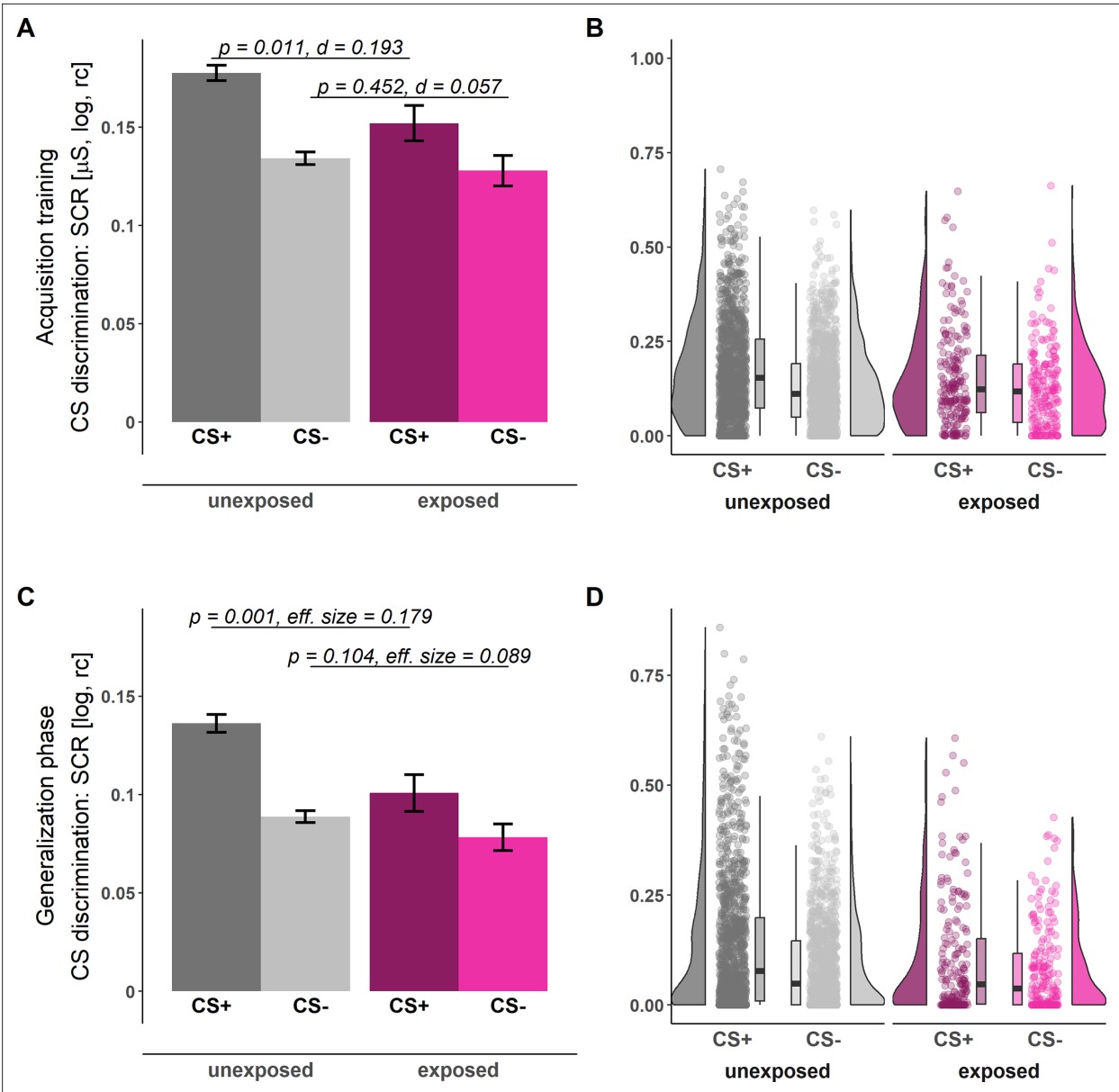

**Figure 2.** Illustration of skin conductance responses (SCRs) during acquisition training (**A–B**) and the generalization phase (**C–D**) for individuals unexposed (gray) and exposed (pink) to childhood adversity separated by stimulus types (CS+: dark shades, CS-: light shades). Barplots (**A and C**) with error bars represent means and standard errors of the means (SEMs) including $n_{unexposed}$ = 1199 and $n_{exposed}$ = 203, respectively. The presented statistical parameters are derived from a two-tailed independent-samples t-test (**A**) and a Yuen independent-samples t-test for trimmed means (**C**). The a priori significance level was set to α = 0.05. Distributions of the data are illustrated in the raincloud plots (**B and D**). Points next to the densities represent the SCRs of each participant as a function of stimulus type averaged across phases. Boxes of boxplots represent the interquartile range (IQR) crossed by the median as a bold line, ends of whiskers represent the minimum/maximum value in the data within the range of $25^{th}/75^{th}$ percentiles ± 1.5 IQR. CS = conditioned stimulus, log = log-transformed, rc = range-corrected.

types (i.e. subscales) are distributed among the different numbers of subscales, see *Appendix 1— figure 5*.

The operationalization of childhood adversity in the context of the specificity model tests the association between exposure to abuse and neglect experiences on conditioned responding statistically independently, while the dimensional model controls for each other's impact (see *Table 2* for details and *Figure 7* for an exemplary illustration of SCRs during acquisition training). Despite these conceptual and operationalizational differences, results are converging. More precisely, no significant effect of exposure to abuse was observed on CS discrimination, the strength of generalization (i.e. LDS), or

general reactivity in any of the outcome measures and in any experimental phase (see *Appendix 1—table 5*; *Appendix 1—table 7*). In contrast, a significant negative association between exposure to neglect and CS discrimination in SCRs was observed during acquisition training (specificity model: F(1, 1400)=6.4, p=0.012, $R^2$=0.005; dimensional model: F(3, 1398)=2.91, p=0.234, $R^2$=0.006), which is contrary to the predictions of the dimensional model, that posits a specific role for abuse but not neglect (*Machlin et al., 2019*; *McLaughlin et al., 2021*). Post hoc tests revealed that in both models, effects were driven by significantly lower SCRs to the CS+ (specificity model: F(1, 1400)=6.13, p=0.013, $R^2$=0.004, dimensional model: ß=–0.004, t(1398)=–1.97, p=0.049, r=–0.07). Within the dimensional model framework, the issue of multicollinearity among predictors (i.e. different childhood adversity types) is frequently discussed (*McLaughlin et al., 2021*; *Smith and Pollak, 2021*). If we apply the rule of thumb of a variance inflation factor (VIF) >10, which is often used in the literature to indicate concerning multicollinearity (e.g. *Hair et al., 1995*; *Mason et al., 1989*; *Neter et al., 1989*), we can assume that multicollinearity was not a concern in our study (abuse: VIF=8.64; neglect: VIF=7.93). However, some authors state that VIFs should not exceed a value of 5 (e.g. *Akinwande et al., 2015*), while others suggest that these rules of thumb are rather arbitrary (*O'brien, 2007*).

Furthermore, the statistical analyses of the specificity model additionally revealed that greater exposure to neglect significantly predicted a generally lower SCR reactivity (F(1, 1400)=4.3, p=0.038, $R^2$=0.003) as well as a lower CS discrimination in contingency ratings during both acquisition training (F(1, 1400)=5.58, p=0.018, $R^2$=0.004) and the generalization test (F(1, 1400)=6.33, p=0.012, $R^2$=0.005; see *Appendix 1—table 6*). These were driven by significantly higher CS- responding in contingency ratings (acquisition training: F(1, 1400)=4.62, p=0.032, $R^2$=0.003; generalization test: F(1, 1400)=8.38, p=0.004, $R^2$=0.006) in individuals exposed to neglect.

To explore the explanatory power of different theories, we exemplarily compared the absolute values of Cohen's d of all exploratory analyses including CS discrimination in SCRs during acquisition training with the absolute values of the Cohen's d confidence intervals of our main analyses. We chose CS discrimination during fear acquisition training for this test, because the most convergent results across theories were observed during this experimental phase. None of the effect sizes from the exploratory analyses (cumulative risk, severity groups: d=0.14; cumulative risk, number of subscales exceeding an at least moderate cut-off: d=0.20; specificity model, abuse: d=0.10; specificity model, neglect: d=0.19; dimensional model: d=0.18) fell outside the confidence intervals of our main results

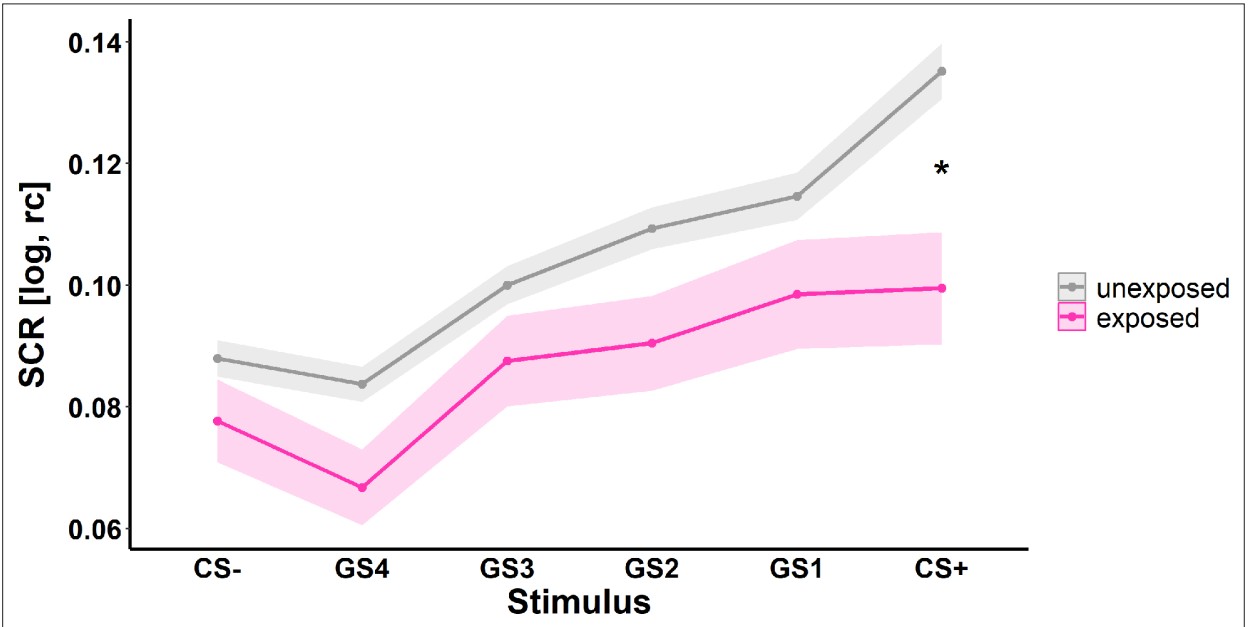

**Figure 3.** Illustration of skin conductance responses (SCRs) to the different stimulus types during the generalization phase (i.e. generalization gradients) for individuals unexposed (gray) and exposed (pink) to childhood adversity. Ribbons represent standard errors of the means (SEMs) including $n_{unexposed}$ = 1199 and $n_{exposed}$ = 203, respectively. CS = conditioned stimulus, GS = generalization stimuli with gradual perceptual similarity to the CS+ and CS-, respectively. log = log-transformed, rc = range-corrected. *p<0.05.

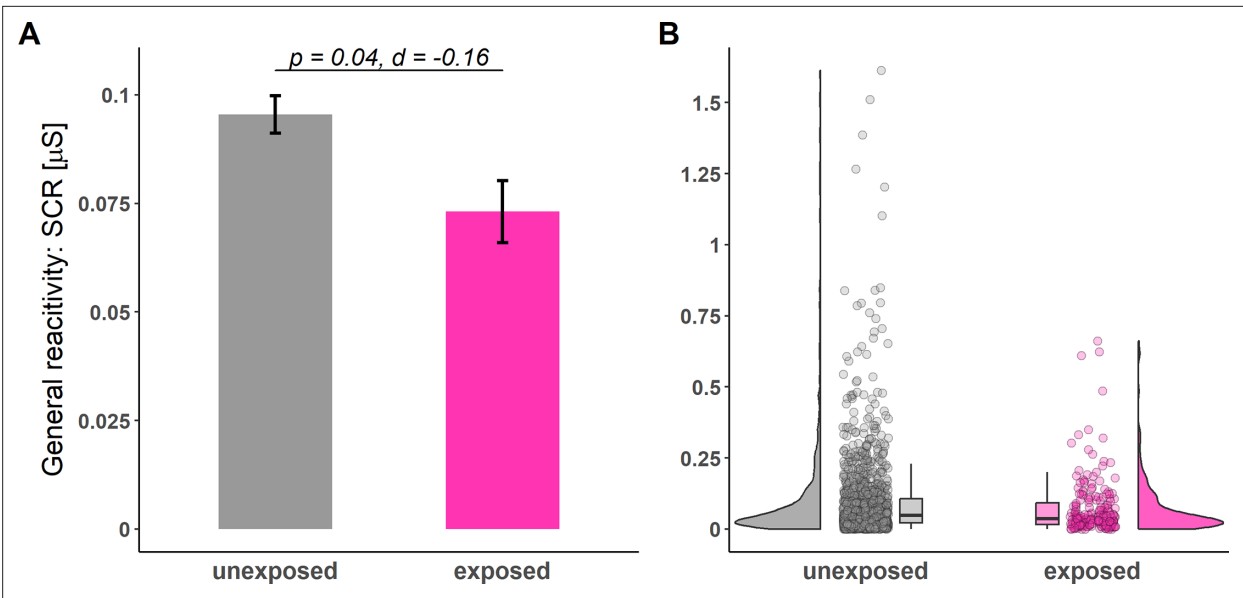

**Figure 4.** Illustration of general reactivity in skin conductance responses (SCRs) across all experimental phases for individuals unexposed (gray) and exposed (pink) to childhood adversity. Barplots (**A**) with error bars represent means and standard errors of the means (SEMs) including $n_{unexposed}$ = 1199 and $n_{exposed}$ = 203, respectively. The statistical parameters presented in **A** are derived from a two-tailed independent-samples t-test. The a priori significance level was set to $\alpha$ = 0.05. Distributions of the data are illustrated in the raincloud plots (**B**). Points next to the densities represent the general reactivity of each participant averaged across all phases. Boxes of boxplots represent the interquartile range interquartile range (IQR) crossed by the median as a bold line, ends of whiskers represent the minimum/maximum value in the data within the range of 25th/75th percentiles ± 1.5 IQR.

(i.e. an at least moderate childhood adversity exposure: [0.03; 0.33]). Hence, we found no evidence of differential explanatory strengths among theories.

## Analyses of trait anxiety and depression symptoms

As expected, participants exposed to childhood adversity reported significantly higher trait anxiety and depression levels than unexposed participants (all $p$'s<0.001; see *Table 3* and *Appendix 1—figure 6*). This pattern remained unchanged when childhood adversity was operationalized differently - following the cumulative risk approach, the specificity, and the dimensional model (see the Materials and methods section). These additional analyses all indicated a significant positive relationship between exposure to childhood adversity and trait anxiety as well as depression scores irrespective of the specific operationalization of 'exposure' (see *Appendix 1—figure 7*).

CS discrimination during acquisition training and the generalization phase, generalization gradients, and general reactivity in SCRs were unrelated to trait anxiety and depression scores in this sample with the exception of a significant association between depression scores and CS discrimination during fear acquisition training (see *Appendix 1—table 8*). More precisely, a very small but significant negative correlation was observed indicating that high levels of depression were associated with reduced levels of CS discrimination ($r$=–0.057, p=0.033). The correlation between trait anxiety levels and CS discrimination during fear acquisition training was not statistically significant but on a descriptive level, high trait anxiety scores were also linked to lower CS discrimination scores ($r$=–0.05, p=0.06) although we highlight that this should not be overinterpreted in light of the large sample. However, both correlations (i.e. CS discrimination during fear acquisition training and trait anxiety as well as depression, respectively) did not statistically differ from each other (z=0.303, p=0.762, *Dunn and Clark, 1969*). Interestingly, and consistent with our results showing that the relationship between childhood adversity and CS discrimination was mainly driven by significantly lower CS+ responses in exposed individuals, trait anxiety and depression scores were significantly associated with SCRs to the CS+, but not to the CS- during acquisition training (see *Appendix 1— table 8*).

**Table 3.** Descriptive information on the subsamples being exposed or unexposed to childhood adversity.

| Variable | Exposed | Unexposed | Statistics |
|---|---|---|---|
| N | 203 (14%) | 1199 (86%) | $X^2(1)=707.57$, p<0.001 |
| Female/Male | 124 (61%) / 79 (39%) | 721 (61%) / 478 (39%) | $X^2(1)=0.03$, p=0.858 |
| Age (M/SD) | 26.80 (6.99) | 25.14 (5.50) | $t(246.1)=-3.21$, p<0.001, d=0.29 |
| STAI-T sum (M/SD) | 38.73 (9.52) | 34.04 (7.83) | $t(250.4)=-6.65$, p<0.001, d=0.58 |
| ADS-K sum (M/SD) | 8.71 (6.31) | 6.69 (5.70) | $t(261)=-4.28$, p<0.001, d=0.35 |

Note. STAI-T = State-Trait Anxiety Inventory, Trait scale (**Spielberger, 1983**), ADS-K = Allgemeine Depressionsskala - Kurzform (short version of the Center for Epidemiological Studies-Depression Scale, CES-D; **Hautzinger and Bailer, 1993**). Individuals were classified as exposed to childhood adversity if at least one subscale met the published cut-off (**Bernstein and Fink, 1998**; **Häuser et al., 2011**) for an at least moderate exposure (i.e. emotional abuse ≥13, physical abuse ≥10, sexual abuse ≥8, emotional neglect ≥15, physical neglect ≥10).

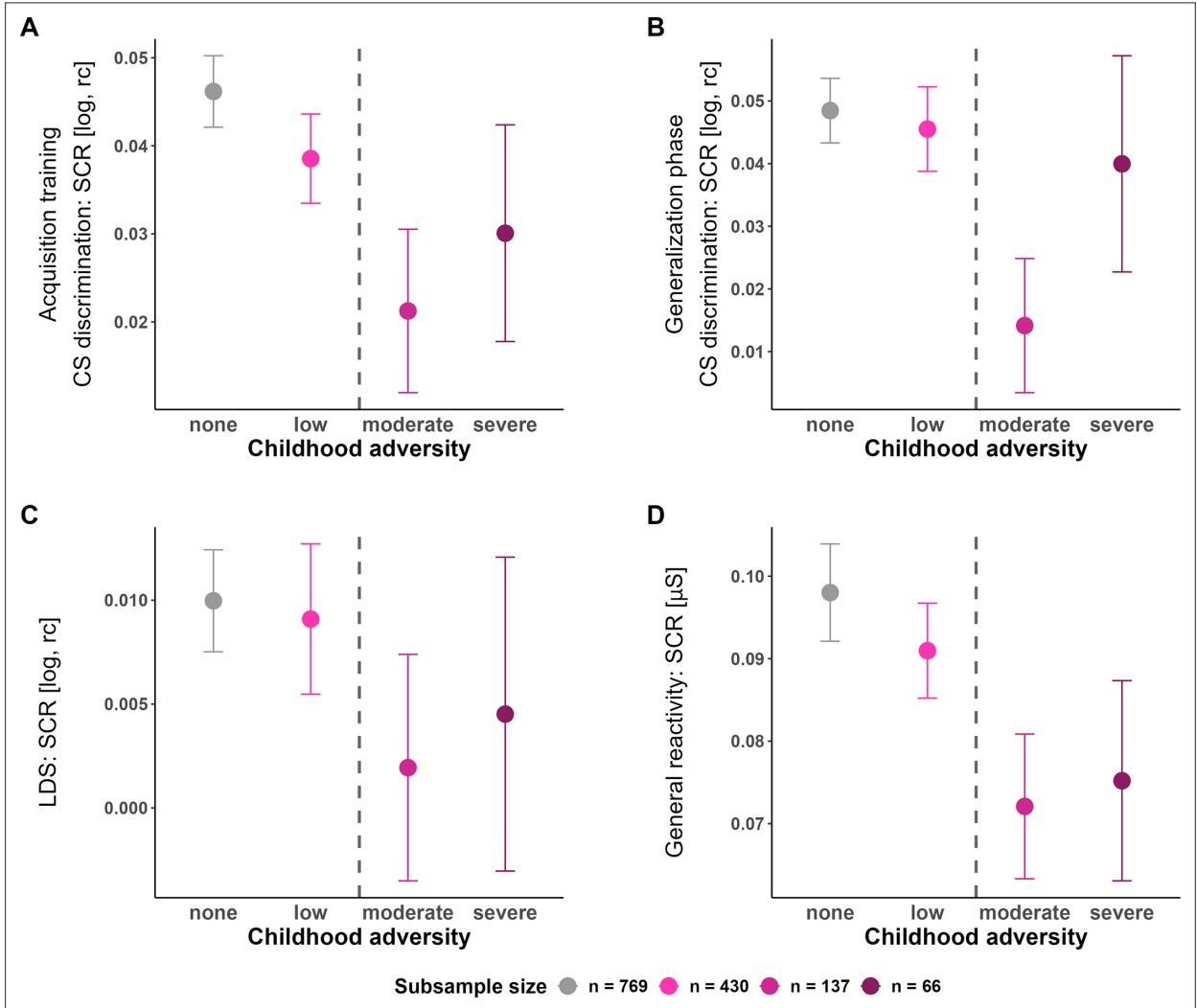

**Figure 5.** Means and standard errors of the mean (SEMs) of conditioned stimulus (CS) discrimination in skin conductance responses (SCRs) during acquisition training (**A**) and the generalization phase (**B**), Linear deviation score (LDS) (**C**), and general reactivity in SCRs (**D**) for the four Childhood Trauma Questionnaire (CTQ) severity groups, respectively. The dashed line indicates the moderate CTQ cut-off frequently used in the literature and hence also employed in our main analyses: On a descriptive level, CS discrimination in SCRs during acquisition training and generalization test, as well as the strength of generalization (i.e. LDS) and the general reactivity are lower in all groups exposed to childhood adversity at an at least moderate level as compared to those with no or low exposure - which corresponds to the main analyses (see above). log = log-transformed, rc = range-corrected.

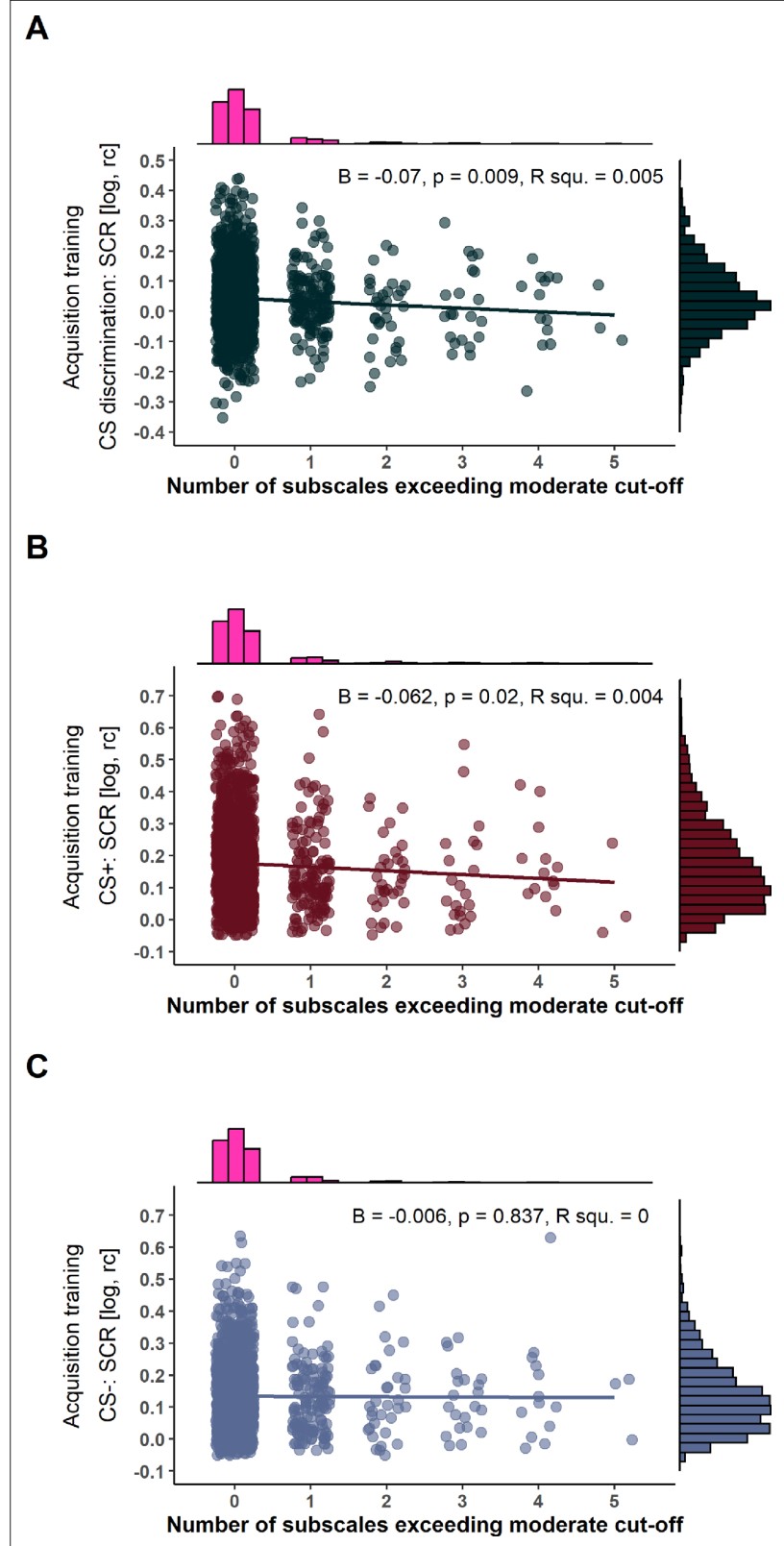

**Figure 6.** Scatterplots with marginal densities illustrating the associations between the number of Childhood Trauma Questionnaire (CTQ) subscales exceeding a moderate or higher cut-off (*Häuser et al., 2011*) and conditioned stimulus (CS) discrimination in skin conductance responses (SCRs) (**A**) as well as SCRs to the CS+ (**B**) and CS- (**C**) during acquisition training. log = log-transformed, rc = range-corrected.

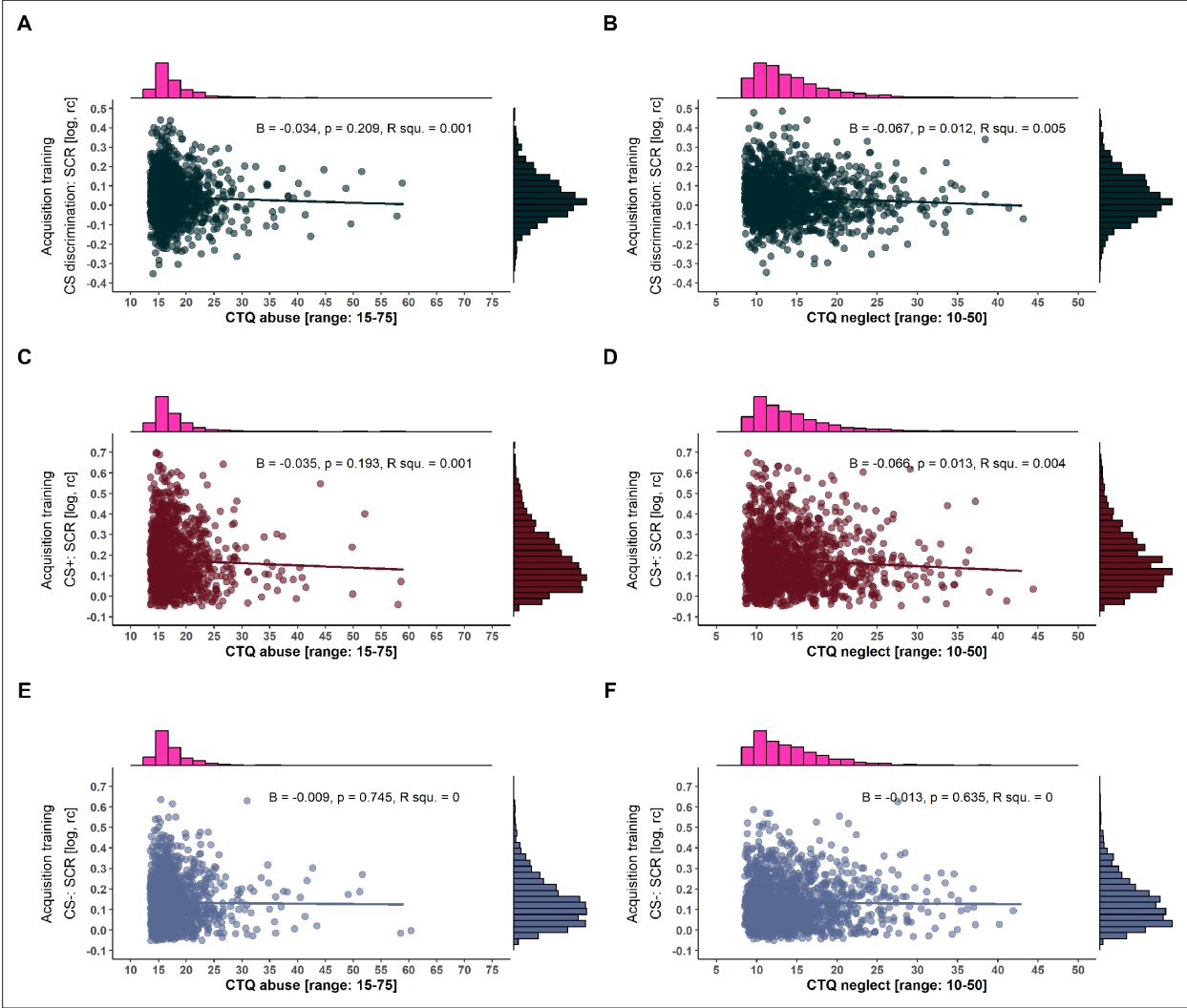

**Figure 7.** Scatterplots with marginal densities illustrating the associations between Childhood Trauma Questionnaire (CTQ) composite scores of abuse (left panel) and neglect (right panel) and conditioned stimulus (CS) discrimination in skin conductance responses (SCRs) (**A and B**) as well as SCRs to the CS+ (**C and D**) and CS- (**E and F**) during acquisition training. Note that the different ranges of CTQ composite scores result from summing up two and three subscales for the neglect and abuse composite scores, respectively (see also *Table 1* for more details). log = log-transformed, rc = range-corrected, R squ.=R squared.

## Discussion

The objective of this study was to examine the relationship between the exposure to childhood adversity and conditioned responding using a large community sample of healthy participants. This relationship might represent a potential mechanistic route linking experience-dependent plasticity in the nervous system and behavior related to risk of and resilience to psychopathology. In additional exploratory analyses, we examined these associations through different approaches by translating key theories in the literature into statistical models. In line with the conclusion of a recent systematic literature review (*Ruge et al., 2024*), individuals exposed to (an at least moderate level of) childhood adversity exhibited reduced CS discrimination in SCRs during both acquisition training and the generalization phase compared to those classified as unexposed (i.e. no or low exposure). Generalization gradients themselves were, however, comparable between exposed and unexposed individuals.

The systematic literature search by *Ruge et al., 2024* revealed that the pattern of decreased CS discrimination, driven primarily by reduced CS+ responding, was observed despite substantial heterogeneity in childhood adversity assessment and operationalization, and despite differences in the experimental paradigms. Although both individuals without mental disorders exposed to childhood

adversity and patients suffering from anxiety- and stress-related disorders (e.g. *Duits et al., 2015*) show reduced CS discrimination, it is striking that the response pattern of individuals exposed to childhood adversity (i.e. reduced responding to the CS+) is remarkably different from what is typically observed in patients (i.e. enhanced responding to the CS-). It should be noted, however, that childhood adversity exposure status was not considered in this meta-analysis (*Duits et al., 2015*). As exposure to childhood adversity represents a particularly strong risk factor for the development of later psychopathology, these seemingly contrary findings warrant an explanation. In this context, it is important to note that all individuals included in the present study were mentally healthy - at least up to the assessment. Hence, it may be an obvious explanation that reduced CS discrimination driven by decreased CS+ responding may represent a resilience rather than a risk factor because individuals exposed to childhood adversity in our sample are mentally healthy despite being exposed to a strong risk factor.

In fact, there is substantial heterogeneity in individual trajectories and profiles in the aftermath of such exposures in humans and rodents (*Russo et al., 2012*). While some individuals remain resilient despite exposure, others develop psychopathological conditions (*Galea et al., 2005*). Consequently, the sample of the present study may represent a specific subsample of exposed individuals who are developing along a resilient trajectory. Thus, it can be speculated that reduced physiological reactivity to a signal of threat (e.g. CS+) may protect the individual from overwhelming physiological and/or emotional responses to potentially recurrent threats (for a discussion, see *Ruge et al., 2024*). Similar concepts have been proposed as 'emotional numbing' in post-traumatic stress disorders (for a review, see e.g. *Litz and Gray, 2002*).

While this seems a plausible theoretical explanation, decreased CS discrimination driven by reduced CS+ responding is observed rather consistently and most importantly irrespective of whether the investigated samples were healthy, at risk, or included patients (*Ruge et al., 2024*). Thus, this response pattern which was also observed in our study might be a specific characteristic of childhood adversity exposure distinct from the response pattern generally observed in patients suffering from anxiety- and stress-related disorders (i.e. increased responding to the CS-) - even though individuals exposed to childhood adversity in this sample indeed showed significantly higher anxiety and depression despite being free of any categorical diagnoses (see Appendix 1) which was also previously reported by e.g., *Spinhoven et al., 2011* and *Kuhn et al., 2016*. Interestingly, in our study, trait anxiety and depression scores were mostly unrelated to SCRs, defined by CS discrimination and generalization gradients based on SCRs as well as general SCR reactivity, with the exception of a significant - albeit minute - relationship between CS discrimination during acquisition training and depression scores (see above). Although reported associations in the literature are heterogeneous (*Lonsdorf et al., 2017*), we may speculate that they may be mediated by childhood adversity. We conducted additional mediation analyses (data not shown) which, however, did not support this hypothesis. As the potential links between reduced CS discrimination in individuals exposed to childhood adversity and the developmental trajectories of psychopathological symptoms are still not fully understood, future work should investigate these further in - ideally - prospective studies.

In addition to reduced CS discrimination in SCRs, a generally blunted electrodermal responding was observed, which may, however, be mainly driven by substantially reduced CS+ responses. Yet, it is noteworthy that reduced skin conductance in children exposed to childhood adversity was also observed during other tasks such as attention regulation during interpersonal conflict (*Pollak et al., 2005*), or passively viewing slides with emotional or cognitive content (*Carrey et al., 1995*), whereas other studies did not find such an association (*Ben-Amitay et al., 2016*). Moreover, in various threat-related studies, also enhanced responding or no significant differences were observed across outcome measures (*Estrada et al., 2020*; *Huskey et al., 2022*; *Jovanovic et al., 2009*; *Jovanovic et al., 2022*; *Kreutzer and Gorka, 2021*; *Lis et al., 2020*; *Pole et al., 2007*; *Rowland et al., 2022*; *Thome et al., 2018*; *Young et al., 2018*). While generally blunted responding might be particularly related to decreased CS+ responding in the present study, differences in general reactivity need to be taken into account for data analyses and interpretation - in particular as the exclusion of physiological non-responders or so-called 'non-learners' (i.e. individuals not showing a minimum discrimination score between SCRs to the CS+ and CS-) has been common in the field until recently (for a critical discussion, see *Lonsdorf et al., 2019*). Future work should also investigate reactivity to the unconditioned stimulus, which was not implemented here and may shed light on potential differences in

general reactivity unaffected by associative learning processes (see e.g. *Harnett et al., 2019*; *Machlin et al., 2019*).

Contrary to the association between CS discrimination as well as general (electrodermal) reactivity and exposure to childhood adversity, no such relationship was found for generalization gradients. In a subsample of this study, it was previously observed that fear generalization phenotypes explained less variance as compared to CS discrimination and general reactivity (*Stegmann et al., 2019*), and CS discrimination as well as general reactivity but not fear generalization predicted increases in anxiety and depression scores during the COVID-19 pandemic (*Imholze et al., 2023*). The lack of associations with fear generalization measures (i.e. LDS) may be specific to the paradigm and sample used in these studies, but it may also be an interesting lead for future work to disentangle the relationship between CS discrimination, general reactivity, and generalization gradients, as they have been suggested to be interrelated (*Imholze et al., 2023*; *Stegmann et al., 2019*).

In sum, the current results converge with the literature in identifying reduced CS discrimination and decreased CS+ responding as key characteristics in individuals exposed to childhood adversity. As highlighted recently (see *Koppold et al., 2023*; *Ruge et al., 2024*), the various operationalizations of childhood adversity as well as general trauma (*Karstoft and Armour, 2023*) represent a challenge for integrating the current results into the existing literature. Hence, future studies should focus on in-depth phenotyping (*Ruge et al., 2024*), an elaborate classification of adversity subtypes (*Pollak and Smith, 2021*), and methodological considerations (*Ruge et al., 2024*). Besides optimizing the operationalization of childhood adversity, there are also initiatives to advance the field in data processing and analysis, such as developing methods to address multicollinearity in childhood adversity data (*Brieant et al., 2024*).

Several proposed (verbal) theories describe the association between (specific) childhood adversity types and behavioral as well as physiological consequences differently and there is currently a heated debate rather than consensus on this issue (*McLaughlin et al., 2021*; *Pollak and Smith, 2021*; *Smith and Pollak, 2021*). In the field, most often a dichotomization in exposed vs. unexposed individuals is used (for a review, see *Ruge et al., 2024*). We adopted this typical approach of an at least moderate exposure cut-off from the literature for our main analyses, despite the well-known statistical disadvantages of artificially dichotomizing variables that are (presumably) dimensional in nature (*Cohen, 1983*). It is noteworthy, however, that this cut-off appears to map rather well onto the psychophysiological response patterns observed here (see *Figure 5*). More precisely, our exploratory results of applying different exposure cut-offs (low, moderate, severe, no exposure) seem to indicate that indeed a moderate exposure level is 'required' for the manifestation of physiological differences, suggesting that childhood adversity exposure may not have a linear or cumulative effect.

Of note, comparing individuals exposed vs. unexposed to an at least moderate level of childhood adversity is not derived from any of the existing theories, but rather from practices in the literature (see *Ruge et al., 2024*). For this reason, we aimed at an exploratory translation of key (verbal) theories into statistical models (see *Table 1*). Several important topical and methodological take-home messages can be drawn from this endeavor: First, the translation of these verbal theories into precise statistical tests proved to be a rather challenging task paved by operationalizational ambiguity. We have collected some key challenges in *Table 1* and conclude that current verbal theories are, at least to a certain degree, ill-defined, as our attempt has disclosed a multiverse of different, equally plausible ways to test them - even though we provide only a limited number of exemplary tests. Second, despite these challenges, the results of most tests converged in identifying an effect of childhood adversity on reduced CS discrimination in SCRs during acquisition training, which is reassuring when aiming to integrate results based on different operationalizations. Third, none of the theories appears to be explanatorily superior. Fourth, our results are not in line with predictions of the dimensional model (*Machlin et al., 2019*; *McLaughlin et al., 2021*) which posits a specific association between exposure to threat- but not deprivation-related childhood adversity and fear conditioning performance. If anything, our results point in the opposite direction.

Taken together, neither considering childhood adversity as a broad category, nor different subtypes have consistently shown to strongly map onto biological mechanisms (for an in-depth discussion, see *Smith and Pollak, 2021*). Hence, even though it is currently the dominant view in the field, that considering the potentially distinct effects of dissociable adversity types holds promise to provide mechanistic insights into how childhood adversity becomes biologically embedded (*Berens et al.,*

2017; *Kuhlman et al., 2017*; *Smith and Pollak, 2021*), we emphasize the urgent need for additional exploration, refinement, and testing of current theories. This is particularly important in light of diverging evidence pointing towards different conclusions.

Some limitations of this work are worth noting: First, despite our observation of significant associations between exposure to childhood adversity and fear conditioning performance in a large sample, it should be noted that effect sizes were small. Second, we cannot provide a comparison of potential group differences in unconditioned responding to the US. This is, however, important as this comparison may explain group differences in conditioned responding - a mechanism that remains unexplored to date (*Ruge et al., 2024*). Third, the use of the CTQ, which is the most commonly used questionnaire in the field (see *Ruge et al., 2024*), comes with a number of disadvantages. Most prominently, the CTQ focuses exclusively on the presence or absence of exposure without consideration of individual and exposure characteristics that have been shown to be of crucial relevance (*Danese and Widom, 2023*; see *Smith and Pollak, 2021*), such as controllability, burdening, exposure severity, duration, and developmental timing. These characteristics are embedded in the framework of the topological approach (*Smith and Pollak, 2021*), another important model linking childhood adversity exposure to negative outcomes, which, however, was not evaluated in the present work. Testing this model requires an extremely large dataset including in-depth phenotyping, which was not available here, but may be an important avenue for future work. Fourth, across all theories, significant effects of childhood adversity have been shown primarily on physiological reactivity (i.e. SCR). Whether these findings are specific to SCRs or might generalize to other physiological outcome measures such as fear-potentiated startle, heart rate, or local changes in neural activation, remains an open question for future studies.

In sum, when ultimately aiming to understand the impact of exposure to adversity on the development of psychopathological symptoms (*Anda et al., 2006*; *Felitti, 2002*; *Gilbert et al., 2009*; *Green et al., 2010*; *Heim and Nemeroff, 2001*; *McLaughlin et al., 2012*; *Moffitt et al., 2007*; *Teicher et al., 2022*), it is crucial to understand the biological mechanisms through which exposure to adversity 'gets under the skin.' To achieve this, emotional-associative learning can serve as a prime translational model for fear and anxiety disorders: One plausible mechanism is the ability to distinguish threat from safety, which is key to an individual's ability to dynamically adapt to changing environmental demands (*Craske et al., 2012*; *Vervliet et al., 2013*) - an ability that appears to be impaired in individuals with a history of childhood adversity. This mechanism is of particular relevance to the development of stress- and anxiety-related psychopathology, as the identification of risk but also resilience factors following exposure to childhood adversity is essential for the development of effective intervention and prevention programs.

## Materials and methods

### Participants

In total, 1678 healthy participants (age$_M$=25.26 years, age$_{SD}$=5.58 years, female=60.10%, male=39.30%) were recruited in a multi-centric study at the Universities of Münster, Würzburg, and Hamburg, Germany (SFB TRR58). Data from parts of the Würzburg sample have been reported previously (*Herzog et al., 2021*; *Imholze et al., 2023*; *Schiele et al., 2020*; *Schiele et al., 2016a*; *Schiele et al., 2016b*; *Stegmann et al., 2019*). These previous reports, also those focusing on experimental fear conditioning (*Schiele et al., 2016a*; *Stegmann et al., 2019*), addressed, however, research questions different from the ones investigated here (see also Appendix 1 for details). The study was approved by the local ethics committees of the three Universities (Münster: 2016–131-b-S, Ethics Committee Westfalen-Lippe; Würzburg: Votum 07/08, Ethics Committee of the Medical Faculty of the University of Würzburg; Hamburg: PV2755, Ethics Committee of the General Medical Council Hamburg) and was conducted in agreement with the Declaration of Helsinki. Current and/or lifetime diagnosis of DSM-IV mental Axis-I disorders, as assessed by the German version of the Mini International Psychiatric Interview (*Sheehan et al., 1998*), led to exclusion from the study (see Appendix 1 for additional exclusion criteria). All participants provided written informed consent and received 50 € as compensation.

A reduced number of 1402 participants (age$_M$=25.38 years, age$_{SD}$=5.76 years, female=60.30%, male=39.70%) were included in the statistical analyses because 276 participants were excluded due to missing data (CTQ: n=21, ratings: n=78, SCRs: n=182), for technical reasons, and due to deviating

from the study protocol. Five participants had missing CTQ and missing SCR data. Thus, the sum of exclusions in specific outcome measures does not add up to the total number of exclusions. We did not exclude physiological SCR non-responders or non-learners, as this procedure has been shown to induce bias through predominantly excluding specific subpopulations (e.g. high trait anxiety), which may be particularly prevalent in individuals exposed to childhood adversity (*Lonsdorf et al., 2019*). See *Table 3* and Appendix 1 for additional sample information including trait anxiety and depression scores (see *Appendix 1—figures 6 and 7*), zero-order correlations (Pearson's correlation coefficient) between trait anxiety, depression scores, and childhood adversity (see *Appendix 1—figure 1*) as well as information on SES (see *Appendix 1—figure 2*).

## Procedure

### Fear conditioning and generalization paradigm

Participants underwent a fear conditioning and generalization paradigm which was adapted from *Lau et al., 2008* and described previously in detail (*Herzog et al., 2021*; *Schiele et al., 2016a*; *Stegmann et al., 2019*). Details are also provided in brief in the Appendix 1 (see also *Appendix 1—figure 3*).

### Ratings

At the end of each experimental phase (habituation, acquisition training, and generalization) as well as after half of the total acquisition and generalization trials, participants provided ratings of the faces with regards to valence, arousal (9-point Likert-scales; from 1=very unpleasant/very calm to 9=very pleasant/very arousing) and US contingencies (11-point Likert-scale; from 0 to 100% in 10% increments). As the US did not occur during the habituation phase, contingency ratings were not provided after this phase. For reasons of comparability, valence ratings were inverted.

## Physiological data recordings and processing

Skin conductance was recorded continuously using Brainproducts V-Amp-16 and Vision Recorder software (Brainproducts, Gilching, Germany) at a sampling rate of 1000 Hz from the non-dominant hand (thenar and hypothenar eminences) using two Ag/AgCl electrodes. Data were analyzed offline using BrainVision Analyzer 2 software (Brainproducts, Gilching, Germany). The signal was filtered offline with a high cut-off filter of 1 Hz and a notch filter of 50 Hz. Amplitudes of SCRs were quantified by using the Trough-to-peak (TTP) approach. According to published guidelines (*Boucsein et al., 2012*), the response onset was defined as between 900–4000 ms after stimulus onset and the peak between 2000–6000 ms after stimulus onset. A minimum response criterion of 0.02 µS was applied, with lower individual responses scored as zero (i.e. non-responses). Note that previous work using this sample (*Schiele et al., 2016a*; *Stegmann et al., 2019*) had used square-root transformations but we decided to employ a log-transformation and range-correction (i.e. dividing each SCR by the maximum SCR per participant). We used log-transformation and range-correction for SCR data because these transformations are standard practice in our laboratory and we strive for methodological consistency across different projects (e.g. *Ehlers et al., 2020*; *Kuhn et al., 2016*; *Scharfenort et al., 2016*; *Sjouwerman and Lonsdorf, 2020*; *Sjouwerman et al., 2015*). Additionally, log-transformed and range-corrected data are generally assumed to approximate a normal distribution more closely and exhibit lower error variance, which leads to larger effect sizes (*Lykken, 1972*; *Lykken and Venables, 1971*; *Sjouwerman et al., 2022*). Additionally, on a descriptive level, this combination of transformations appears to offer greater reliability compared to using raw data alone (*Klingelhöfer-Jens et al., 2022*).

## Psychometric assessment

Participants completed a computerized battery of questionnaires (for a full list, see *Stegmann et al., 2019*) prior to the experiment including a questionnaire with general questions asking, for example, about the SES, the German versions of the trait version of the State-Trait Anxiety Inventory (STAI-T, *Spielberger, 1983*), the CTQ-SF (*Bernstein et al., 2003*; *Wingenfeld et al., 2010*) and the short version of the Center for Epidemiological Studies-Depression Scale (CES-D, in Germany: Allgemeine Depressionsskala - Kurzform, ADS-K; *Hautzinger and Bailer, 1993*). The CTQ contains 28 items for the retrospective assessment of childhood adversity across five subscales (emotional, physical, and sexual abuse, as well as emotional and physical neglect; for internal consistency, see Appendix 1), and

a control scale. The STAI-T consists of 20 items addressing trait anxiety (*Laux and Spielberger, 1981*; *Spielberger, 1983*), and the ADS-K includes 15 items assessing depressiveness during the past 7 d.

## Operationalization of 'exposure'

We implemented different approaches to operationalize exposure to childhood adversity in the main analyses and exploratory analyses (see *Table 1*). In the main analyses, we followed the approach most commonly employed in the field of research on childhood adversity and threat learning - using the moderate exposure cut-off of the CTQ (for a recent review see *Ruge et al., 2024*). In addition, the heterogeneous operationalizations of classifying individuals into exposed and unexposed to childhood adversity in the literature (*Koppold et al., 2023*; *Ruge et al., 2024*) hampers comparison across studies and hence cumulative knowledge generation. Therefore, we also provide exploratory analyses (see below) in which we employ different operationalizations of childhood adversity exposure.

## Statistical analyses

Manipulation checks were performed to test for successful fear acquisition and generalization (for more details, see Appendix 1). Following previous studies (*Imholze et al., 2023*; *Stegmann et al., 2019*), we calculated three different outcomes for each participant for SCRs and ratings: CS discrimination (for acquisition training and the generalization phase), the linear deviation score (LDS; only for the generalization phase) as an index of the linearity of the generalization gradient (*Kaczkurkin et al., 2017*), and the general reactivity (across all phases including habituation, acquisition training and the generalization phase). CS discrimination was calculated by separately averaging responses to CS+ and CS- across trials (except the first acquisition trial) and subtracting averaged CS- responses from averaged CS+ responses. The first acquisition trial was excluded as no learning could possibly have taken place due to the delay conditioning paradigm. The LDS was calculated by subtracting the mean responses to all GSs from the mean responses to both CSs during the generalization phase. To calculate the general reactivity in SCRs and ratings, trials were averaged across all stimuli (CSs and GSs) and phases (i.e. habituation, acquisition training, and generalization phase). Note that raw SCRs were used for analyses of general physiological reactivity.

CS discrimination during acquisition training and the generalization phase, LDS, and general reactivity were compared between participants who were exposed and unexposed to childhood adversity by using two-tailed independent-samples t-tests. For CS discrimination in SCRs, a two-way mixed ANOVA was conducted to examine the effect of childhood adversity exposure on responses to the CS+ and CS- by including CS type and childhood adversity exposure as independent variables. As the interaction between CS type and childhood adversity exposure was statistically significant, post hoc two-tailed paired t-tests were used to compare SCRs between CS+ and CS- within each group and independent-samples t-tests to contrast responses to each CS between exposed and unexposed participants.

## Exploratory analyses

Additionally, the different ways of classifying individuals as exposed or unexposed to childhood adversity in the literature (*Koppold et al., 2023*; for discussion see *Ruge et al., 2024*) hinder comparison across studies and hence cumulative knowledge generation. Therefore, we also conducted exploratory analyses using different approaches to operationalize exposure to childhood adversity (see *Table 1* for details). Note that no correction for alpha inflation was applied in these analyses, given their exploratory nature. To compare the explanatory strengths of the included theories, all effect sizes from the exploratory tests were converted to the absolute value of Cohen's d as the direction is not relevant in this context. When their value fell outside the confidence intervals of the effect sizes of the main analysis (*LeBel et al., 2018*), this was inferred as meaningful differences in explanatory strengths.

## Analyses of trait anxiety and depression symptoms

To further characterize our sample, we compared individuals being unexposed to those exposed to childhood adversity on trait anxiety and depression scores by using Welch's tests due to unequal variances.

On the request of a reviewer, we additionally investigated the association of childhood adversity as operationalized by the different models used in our explanatory analyses (i.e. cumulative risk, specificity, and dimensional model) and trait anxiety as well as depression scores (see *Appendix 1—figure 7*). By using STAI-T and ADS-K scores as independent variables, we calculated (a) a comparison of conditioned responding of the four severity groups (i.e. no, low, moderate, severe exposure to childhood adversity) using one-way ANVOAs and the association with the number of sub-scales exceeding an at least moderate cut-off in simple linear regression models for the implementation of the cumulative risk model, and (b) the association with the CTQ abuse and neglect composite scores in separate linear regression models for the implementation of the specificity/dimensional models. On request of the reviewer, we also calculated the Pearson correlation between trait anxiety (i.e. STAI-T scores), depression scores (i.e. ADS-K scores), and conditioned responding in SCRs (see *Appendix 1—table 8*).

In statistical procedures where the assumption of homogeneity of variance was not met, Welch's tests, robust trimmed means ANOVAs (*Mair and Wilcox, 2020*), and regressions with robust standard errors using the HC3 estimator (*Hayes and Cai, 2007*) were calculated instead of t-tests, ANOVAs and regressions, respectively. Note that for robust mixed ANOVAs, the WRS2 package in R (*Mair and Wilcox, 2020*) does not provide an effect size. In the main analyses, post hoc t-tests or Welch's tests were corrected for multiple comparisons by using the Holm correction. As post hoc tests for robust ANOVAs, Yuen independent samples t-test for trimmed means were calculated including the explanatory measure of effect size (values of 0.10, 0.30, and 0.50 represent small, medium, and large effect sizes, respectively; *Mair and Wilcox, 2020*). Even though such rules of thumb have to be interpreted with caution, we provide these benchmarks here as this effect size might be somewhat unknown.

Following previous calls for a stronger focus on measurement reliability (*Cooper et al., 2023*; *Klingelhöfer-Jens et al., 2022*), we also provide information on split-half reliability for SCRs as well as Cronbach's alpha for the CTQ in the Appendix 1. For all statistical analyses described above, the a priori significance level was set to $\alpha$=0.05. For data analysis and visualizations as well as for the creation of the manuscript, we used R (Version 4.1.3; *R Development Core Team, 2022b*) and the R-packages *apa* (*Aust and Barth, 2020*; Version 0.3.3; *Gromer, 2020*), *car* (Version 3.0.10; *Fox and Weisberg, 2019*; *Fox et al., 2020*), *carData* (Version 3.0.4; *Fox et al., 2020*), *chisq.posthoc. test* (Version 0.1.2; *Ebbert, 2019*), *cocor* (Version 1.1.3; *Diedenhofen and Musch, 2015*), *data.table* (Version 1.13.4; *Dowle and Srinivasan, 2020*), *DescTools* (Version 0.99.42; *Andri, 2021*), *dplyr* (Version 1.1.4; *Wickham et al., 2022*), *effectsize* (Version 0.8.8; *Ben-Shachar et al., 2020*), *effsize* (Version 0.8.1; *Torchiano, 2020*), *ez* (Version 4.4.0; *Lawrence, 2016*), *flextable* (Version 0.9.6; *Gohel, 2021*), *forcats* (Version 0.5.0; *Wickham, 2020*), *foreign* (Version 0.8.82; *R Development Core Team, 2022a*), *ftExtra* (Version 0.6.4; *Yasumoto, 2023*), *GGally* (Version 2.1.2; *Schloerke et al., 2021*), *ggExtra* (Version 0.10.0; *Attali and Baker, 2022*), *gghalves* (Version 0.1.1; *Tiedemann, 2020*), *ggpattern* (Version 1.0.1; *Fc and Davis, 2022*, ggplot2 authors, 2022), *ggplot2* (Version 3.5.1; *Wickham, 2016*), *ggpubr* (Version 0.4.0; *Kassambara, 2020*), *ggsignif* (Version 0.6.3; *Ahlmann-Eltze and Patil, 2021*), *gridExtra* (Version 2.3; *Auguie, 2017*), *haven* (Version 2.3.1; *Wickham and Miller, 2020*), *here* (Version 1.0.1; *Müller, 2020*), *kableExtra* (Version 1.3.1; *Zhu, 2020*), *knitr* (Version 1.37; *Xie, 2015*), *lm.beta* (Version 1.5.1; *Behrendt, 2014*), *lme4* (Version 1.1.26; *Bates et al., 2015*), *lmerTest* (Version 3.1.3; *Kuznetsova et al., 2017*), *lmtest* (Version 0.9.38; *Zeileis and Hothorn, 2002*), *MatchIt* (Version 4.4.0; *Ho et al., 2011*), *Matrix* (Version 1.4.0; *Bates and Maechler, 2021*), *officedown* (Version 0.2.4; *Gohel and Ross, 2022*), *papaja* (Version 0.1.2; *Aust and Barth, 2020*), *patchwork* (Version 1.2.0; *Pedersen, 2020*), *performance* (Version 0.12.0; *Lüdecke et al., 2021*), *psych* (Version 2.0.9; *Revelle, 2020*), *purrr* (Version 1.0.2; *Henry and Wickham, 2020*), *readr* (Version 2.1.4; *Wickham, 2020*), *reshape2* (Version 1.4.4; *Wickham, 2007*), *rstatix* (Version 0.7.0; *Kassambara, 2021*), *sandwich* (*Zeileis, 2004*; *Zeileis, 2006*; Version 3.0.1; *Zeileis et al., 2020*), *sjPlot* (Version 2.8.16; *Lüdecke, 2024*), *stringr* (Version 1.5.1; *Wickham, 2019*), *tibble* (Version 3.2.1; *Müller and Wickham, 2021*), *tidyr* (Version 1.3.1; *Wickham and Girlich, 2022*), *tidyverse* (Version 1.3.0; *Wickham et al., 2019*), *tinylabels* (Version 0.2.3; *Barth, 2022*), WRS2 (Version 1.1.4; *Mair and Wilcox, 2020*), and *zoo* (Version 1.8.8; *Zeileis and Grothendieck, 2005*).

## Acknowledgements

The authors thank Julia Ruge for critical reviewing. KD, MAS, and PZ are members of the Anxiety Disorders Research Network (ADRN) of the European College of Neuropsychopharmacology (ECNP). This work was supported by the German Research Foundation (DFG) – project number 44541416 – TRR 58 'Fear, Anxiety, Anxiety Disorders,' subproject Z02 to JD, KD, UD, TBL, UL, AR, MR, PP; subproject B01 to PP, subproject B07 to TBL, subproject C02 to KD and JD, subproject C10 to MG.

## Additional information

### Funding

| Funder | Grant reference number | Author |
| --- | --- | --- |
| Deutsche Forschungsgemeinschaft | 44541416 - TRR 58 "Fear, Anxiety, Anxiety Disorders" | Jürgen Deckert Katharina Domschke Udo Dannlowski Ulrike Lueken Andreas Reif Marcel Romanos Paul Pauli Matthias Gamer Tina B Lonsdorf |

The funders had no role in study design, data collection and interpretation, or the decision to submit the work for publication.

### Author contributions

Maren Klingelhöfer-Jens, Conceptualization, Software, Formal analysis, Validation, Visualization, Writing – original draft, Writing – review and editing; Katharina Hutterer, Resources, Data curation, Investigation, Writing – review and editing; Miriam A Schiele, Data curation, Supervision, Investigation, Project administration, Writing – review and editing; Elisabeth J Leehr, Joscha Böhnlein, Jonathan Repple, Investigation, Project administration, Writing – review and editing; Dirk Schümann, Karoline Rosenkranz, Data curation, Investigation; Jürgen Deckert, Katharina Domschke, Udo Dannlowski, Ulrike Lueken, Andreas Reif, Marcel Romanos, Peter Zwanzger, Paul Pauli, Matthias Gamer, Resources, Supervision, Funding acquisition, Project administration, Writing – review and editing; Tina B Lonsdorf, Conceptualization, Resources, Supervision, Funding acquisition, Methodology, Writing – original draft, Project administration, Writing – review and editing

### Author ORCIDs

Maren Klingelhöfer-Jens ⓘ https://orcid.org/0000-0002-5393-7871
Miriam A Schiele ⓘ https://orcid.org/0000-0003-2613-6515
Elisabeth J Leehr ⓘ https://orcid.org/0000-0002-9264-5003
Joscha Böhnlein ⓘ https://orcid.org/0000-0002-9870-5599
Jonathan Repple ⓘ https://orcid.org/0000-0003-1379-9491
Katharina Domschke ⓘ https://orcid.org/0000-0002-2550-9132
Udo Dannlowski ⓘ https://orcid.org/0000-0002-0623-3759
Ulrike Lueken ⓘ https://orcid.org/0000-0003-1564-4012
Andreas Reif ⓘ http://orcid.org/0000-0002-0992-634X
Marcel Romanos ⓘ https://orcid.org/0000-0001-7628-8299
Paul Pauli ⓘ https://orcid.org/0000-0003-0692-6720
Matthias Gamer ⓘ https://orcid.org/0000-0002-9676-9038
Tina B Lonsdorf ⓘ https://orcid.org/0000-0003-1501-4846

### Ethics

The study was approved by the local ethics committees of the three Universities (Münster: 2016-131-b-S, Ethics Committee Westfalen-Lippe; Würzburg: Votum 07/08, Ethics Committee of the Medical Faculty of the University of Würzburg; Hamburg: PV2755, Ethics Committee of the General Medical Council Hamburg) and was conducted in agreement with the Declaration of Helsinki. All participants provided written informed consent.

Reviewer #1 (Public review): https://doi.org/10.7554/eLife.91425.3.sa1

Reviewer #2 (Public review): https://doi.org/10.7554/eLife.91425.3.sa2

Author response https://doi.org/10.7554/eLife.91425.3.sa3

## Additional files

### Supplementary files

MDAR checklist

### Data availability

The data will be made available to editors and reviewers only, as publicly sharing of individual-level data was not included in the informed consent forms. Instead, the forms specified that the data would be published anonymously as a collective dataset. At the time the study was planned, data sharing was not a common practice. Therefore, participants were not asked to consent to individual-level data sharing and were assured that their data would be used exclusively for the purposes specified in the consent forms. This restriction also applies to de-identified and processed versions of the individual-level data. R Markdown files that include the code for all analyses and generate this manuscript are openly available at Zenodo (https://doi.org/10.5281/zenodo.14851004; *Klingelhöfer-Jens et al., 2023*).

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

# Appendix 1

## Supplementary methods and results

### Additional information on the paradigm

In brief, two female faces with neutral expressions served as the CS+ and CS- (see also *Appendix 1—figure 3*; CS duration: 6 s, assignment randomized) and a loud female scream 95 dB, delivered via over-ear headphones, US duration: 1.5 s co-occurring with the CS+ face showing a fearful expression that served as the US. Both faces were presented four times each during habituation and 12 times each during acquisition training. During the generalization phase, both faces were shown 12 times in addition to 12 presentations of each of four generalization stimuli (GS) that were morphs of the CS+ and the CS- faces in steps of 20%. During intertrial intervals (ITIs), a white fixation cross on black background was shown for 9–12 s. The generalization phase consisted of two blocks (6 trials each). Since a discrimination training (*Herzog et al., 2021*) was conducted between blocks for some participants, only the first block of the generalization phase was analyzed here. The reinforcement rates were 83.3%, 50%, and 0% during acquisition training, generalization (i.e. to prevent extinction), and habituation, respectively. Participants were not instructed about the CS-US contingencies and were merely informed they should passively view pictures. The participants were provided with an extinction session at the end of the experiment to guarantee that no lasting conditioning would take place (*Schiele et al., 2016a*).

### Additional information on the study sample

Participants of this study were recruited in a multi-centric collaborative research center 'Fear, anxiety, anxiety disorders' joining forces between the Universities of Hamburg, Würzburg, and Münster, Germany (SFB TRR58). During the second funding period (2013–2016), all three sites recruited a large sample (N~500) in the context of the Z project. All participants underwent the cross-sectional experimental paradigm reported here and were additionally extensively characterized to allow specific subprojects to recruit target sub-populations serving different aims with a focus on molecular genetic, epigenetic, or other research questions (see *Herzog et al., 2021*; *Imholze et al., 2023*; *Schiele et al., 2016a*; *Schiele et al., 2016b*; *Stegmann et al., 2019*). The question on the association of exposure to childhood adversity and recent adversity was part of the primary research question of one subproject led by the senior author of this work (B07, TBL) and was hence a research question of primary interest also for this multicentric project.

Additional exclusion criteria included left-handedness, non-Caucasian descent (as sub-projects focused on genetic analyses, e.g. *Schiele et al., 2016b*), pregnancy, severe medical diseases, intake of illegal drugs, psychoactive medication or excessive consumption of alcohol, nicotine, and caffeine (for details see *Schiele et al., 2016a*).

SES in this sample might be roughly inferred from education level and current occupation status. Significant differences were observed between individuals who were exposed as compared to unexposed to childhood adversity in relation to their school degree ($\chi^2(6)=15.89$, p=0.014). More precisely, significantly less individuals (p=0.006) exposed to childhood adversity as compared to those who were unexposed hold a high school diploma (in Germany: 'Abitur') with (still) unfinished studies. However, no significant differences were observed in terms of employment ($\chi^2(8)=14.60$, p=0.067) or type of occupation. ($\chi^2(10)=13.73$, p=0.185; see *Appendix 1—figure 2*).

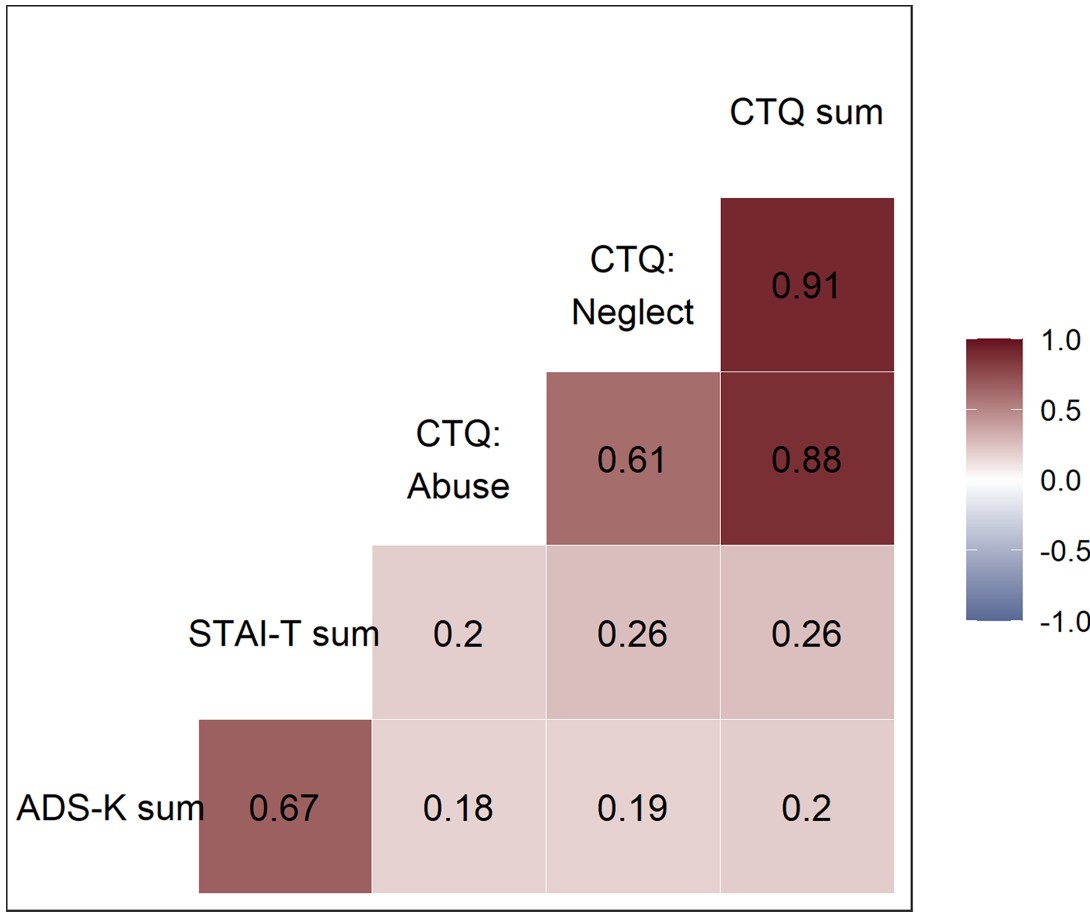

**Appendix 1—figure 1.** Illustration of zero-order correlations (Pearson's correlation coefficient) between relevant sample characteristics: different scores of childhood adversity (CTQ sum, CTQ: Neglect, CTQ: Abuse), trait anxiety (STAI-T sum), and depression (ADS-K sum). All depicted correlations were significant (all *p*'s<0.001). ADS-K=short version of the Center for Epidemiological Studies-Depression Scale (*Hautzinger and Bailer, 1993*), STAI-T=State-Trait Anxiety Inventory, Trait (*Spielberger, 1983*), Abuse and Neglect = composite scores built from childhood trauma questionnaire subscales (CTQ-SF, *Bernstein et al., 2003*; *Wingenfeld et al., 2010*).

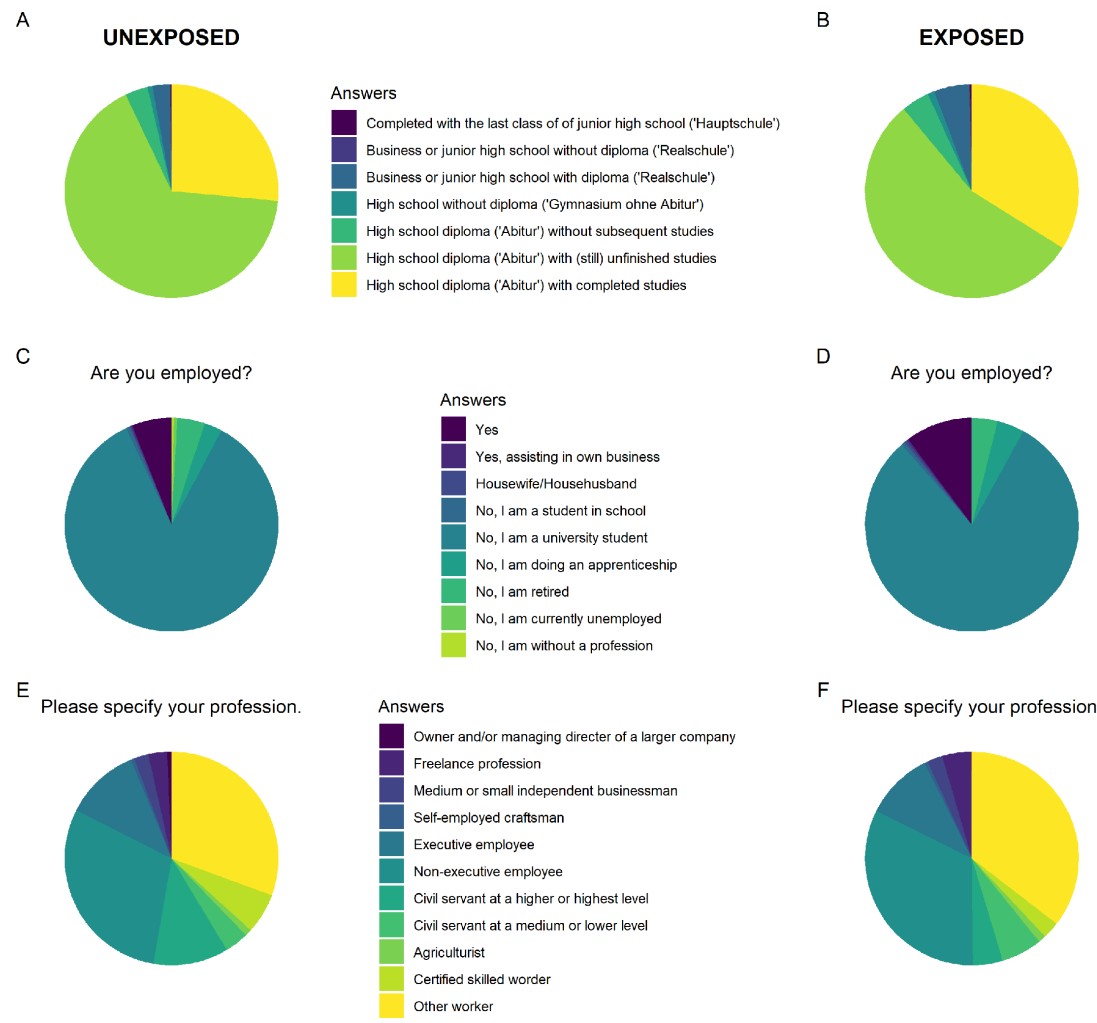

**Appendix 1—figure 2.** Illustration of the socioeconomic status information of individuals unexposed (left) and exposed (right) to childhood adversity (according to an at least moderate childhood adversity cut-off) inferred from questions about the school degree (**A and B**), and the current occupational status (**C and D**) including the type of employment (**E and F**).

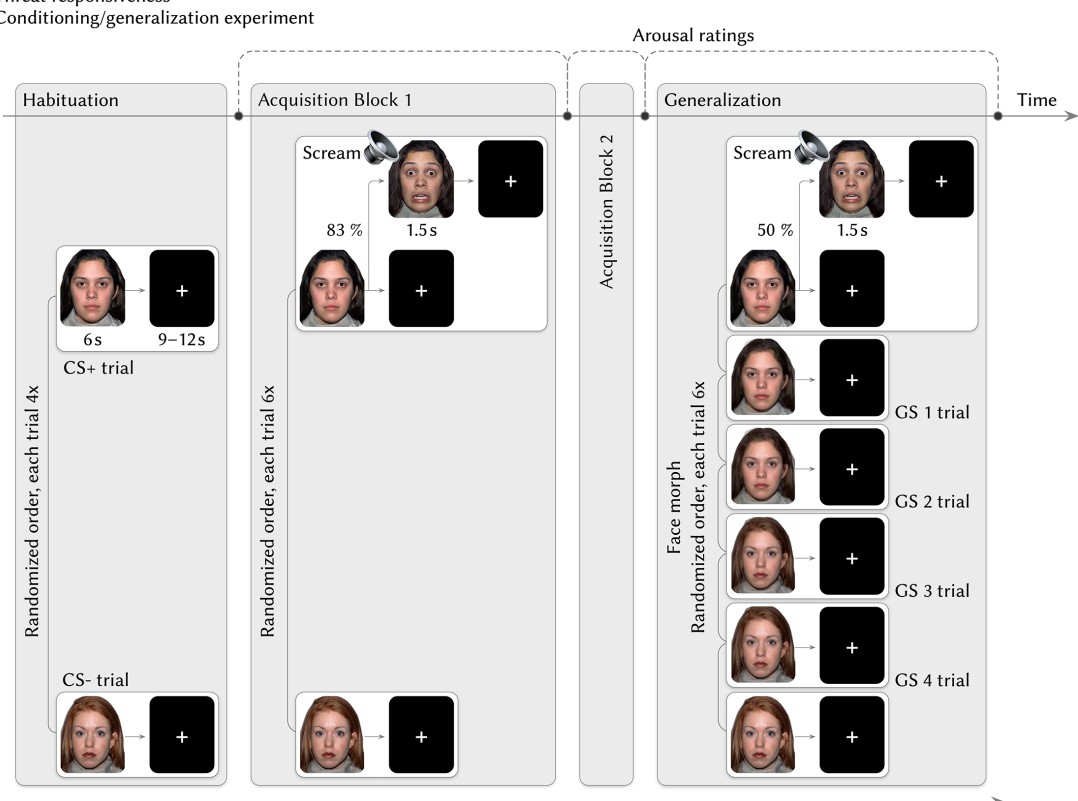

**Appendix 1—figure 3.** This illustration of the study design was created as part of the publication 'Prediction of Changes in Negative Affect During the COVID-19 Pandemic by Experimental Fear Conditioning and Generalization Measures' by *Imholze et al., 2023*; *Figure 1*; https://doi.org/10.1027/2151-2604/a000523, which was distributed as a Hogrefe OpenMind article under the license CC BY 4.0 (https://creativecommons.org/licenses/by/4.0). Please note that the face stimuli used in the actual experiment are different from those shown here. For copyright reasons, we are unable to present the blonde women and have used an image of a redheaded woman as a substitute.

## Supplementary analyses on measurement reliability

Here, we follow recent calls for an increased focus on measurement reliability (*Cooper et al., 2023*; *Klingelhöfer-Jens et al., 2022*; *Zuo et al., 2019*) and report split-half reliability for SCRs for the fear acquisition training and generalization phase. Split-half reliability was calculated by correlating (Pearson's correlation coefficient) averaged odd and even SCR trials (i.e. odd-even approach) for the acquisition and generalization phases separately (see *Appendix 1—table 1*). For acquisition training, the first trial was excluded (*Klingelhöfer-Jens et al., 2022*).

The within-session reliability coefficients observed here for SCRs were comparable to those reported in previous work (*Fredrikson et al., 1993*; see appendix file for an overview of studies in the field in *Klingelhöfer-Jens et al., 2022*) and are, as observed previously, markedly higher for individual stimuli (i.e. CSs and GSs) as opposed to CS discrimination. As the latter is a difference score, this does, however, not come as a surprise (*Infantolino et al., 2018*; *Lynam et al., 2006*; *Moriarity and Alloy, 2021*).

The Cronbach's alpha coefficient for the complete CTQ-SF questionnaire was 0.89, which is consistent with previous studies (alpha=0.94, *Wingenfeld et al., 2010*) and 0.85 and 0.83, for composite scales abuse and neglect, respectively.

**Appendix 1—table 1.** Split-half reliability for skin conductance responses (SCRs).

| Phase | Stimulus type | r | r lower CI | r upper CI |
|---|---|---|---|---|
| | CS+ | 0.802 | 0.783 | 0.820 |
| Acquisition | CS- | 0.735 | 0.710 | 0.758 |
| | CS discr. | 0.322 | 0.274 | 0.368 |
| | CS+ | 0.720 | 0.693 | 0.744 |
| | CS- | 0.470 | 0.429 | 0.510 |
| | CS discr. | 0.346 | 0.299 | 0.391 |
| Generalization | GS1 | 0.656 | 0.625 | 0.684 |
| | GS2 | 0.550 | 0.512 | 0.585 |
| | GS3 | 0.561 | 0.524 | 0.596 |
| | GS4 | 0.288 | 0.240 | 0.336 |

Note. CS = conditioned stimulus, GS = generalization stimulus, CS discr. = CS discrimination.

## Supplementary statistical analyses (main effects of task)

To test for successful manipulation during acquisition training, two-tailed paired t-tests were performed for SCRs and ratings comparing CS+ and CS- responses averaged across trials excluding the first acquisition trial as no learning could possibly have taken place due to the delay conditioning paradigm (*Lonsdorf et al., 2017*). To test for successful manipulation during the generalization phase, a one-way ANOVA with stimulus type as within-subject factor was calculated for SCRs and ratings. Post hoc two-tailed paired t-tests were conducted to test for a gradual increase in responses with increasing similarity to the CS+.

## Supplementary results: Main effects of task

As published previously (*Herzog et al., 2021*; *Schiele et al., 2016a*), successful fear acquisition could be confirmed by significantly stronger responses to the CS+ as compared to CS- in all outcome measures (SCRs: $t(1401)=13.86$, $p<0.001$, $d=0.37$; arousal ratings: $t(1401)=48.26$, $p<0.001$, $d=1.29$; valence ratings: $t(1401)=35.57$, $p<0.001$, $d=0.95$; contingency ratings: $t(1401)=61.00$, $p<0.001$, $d=1.63$). Similarly, ANOVAs indicated significant differences between responses to the CS+, CS-, and GSs during generalization phase (SCRs: $F(4.49, 6284.85)=60.66$, $p<0.001$, $\eta_p^2=0.04$; arousal ratings: $F(3.27, 4578.72)=1031.46$,, $p<0.001$, $\eta_p^2 = 0.42$; valence ratings: $F(3.43, 4809.75)=766.44$, $p<0.001$, $\eta_p^2=.35$; contingency ratings: $F(3.20, 4482.78)=2222.76$, $p<0.001$, $\eta_p^2=0.61$). Post hoc tests yielded that all CSs and GSs differed from each other during generalization phase in all outcome measures (all $p$'s$<0.01$), except for the comparisons of CS- vs. GS4 and GS1 vs. GS2 which were not significant (both $p$'s$=0.12$).

## Supplementary robustness analyses

Robustness analyses were performed for main SCR analyses of CS discrimination during both the acquisition training and generalization phase, as well as the general reactivity by repeating all main analyses as described in the main manuscript with (a) exclusion of physiological non-responders (i.e. participants with only SCRs = 0), (b) exclusion of extreme outliers (i.e. data points +/-3 x interquartile range (IQR) above/below Q3/Q1), (c) study site as a covariate and (d) square root transformed instead of log-transformed and range-corrected SCRs.

The robustness analyses yielded that results did not change substantially i.e., all statistically significant results remained significant [data not shown] - with two exceptions: First, entering site as a covariate in analyses of CS discrimination during acquisition training attenuated the p-value of the childhood adversity exposure effect from p=0.02 to p=0.09. Likewise, the difference in CS discrimination during acquisition training between exposed and unexposed individuals in SCRs dropped from p=0.02 to p=0.06 when SCRs were square-root transformed instead of log-transformed and range-corrected.

**Appendix 1—table 2.** Resu of the repetition of our main analyses using linear mixed models for skin conductance response (SCR) (A), arousal (B), valence (C), and contingency ratings (D).

**A**

| Predictors | CS discrimination SCR during ACQ | | | | CS discrimination SCR during GEN | | | | LDS SCR | | | | General reactivity SCR | | | |
|---|---|---|---|---|---|---|---|---|---|---|---|---|---|---|---|---|
| | Estimates | CI | p | df | Estimates | CI | p | df | Estimates | CI | p | df | Estimates | CI | p | df |
| (Intercept) | 0.18 | −0.03–0.40 | 0.1 | 1367.26 | 0.03 | −0.25–0.30 | 0.841 | 1347.55 | 0.02 | −0.12–0.16 | 0.771 | 1387.92 | 0.16 | −0.10–0.43 | 0.23 | 130.64 |
| Age | 0 | −0.00 – 0.00 | 0.327 | 1388.92 | 0 | −0.00 – 0.00 | 0.334 | 1388.97 | 0 | −0.00 – 0.00 | 0.655 | 1163.35 | 0 | −0.00 – 0.00 | 0.429 | 1386.32 |
| Sex [1] | −0.02 | −0.03 to −0.01 | <0.001 | 1388.99 | 0 | −0.02–0.01 | 0.818 | 1388.99 | 0 | −0.00–0.01 | 0.427 | 1074.14 | 0.01 | −0.00–0.02 | 0.176 | 1386.33 |
| dummy(School level)1 | 0.12 | −0.18–0.41 | 0.448 | 1386 | 0.24 | −0.14–0.63 | 0.22 | 1386 | 0.09 | −0.11–0.28 | 0.382 | 1386.16 | 0.02 | −0.32–0.36 | 0.913 | 1386 |
| dummy(School level)2 | −0.15 | −0.37–0.07 | 0.193 | 1386.7 | 0.03 | −0.26–0.31 | 0.844 | 1386.54 | 0.02 | −0.12–0.17 | 0.753 | 1386.6 | −0.05 | −0.30–0.20 | 0.706 | 1386.03 |
| dummy(School level)3 | −0.09 | −0.33–0.14 | 0.431 | 1386.54 | −0.06 | −0.36–0.24 | 0.699 | 1386.42 | 0.01 | −0.14–0.16 | 0.931 | 1388.1 | −0.06 | −0.33–0.20 | 0.654 | 1386.02 |
| dummy(School level)4 | −0.16 | −0.37–0.06 | 0.153 | 1386.67 | 0.04 | −0.23–0.31 | 0.775 | 1386.52 | 0 | −0.14–0.13 | 0.951 | 1387.75 | −0.03 | −0.27–0.22 | 0.83 | 1386.03 |
| dummy(School level)5 | −0.17 | −0.39–0.05 | 0.132 | 1386.36 | 0.05 | −0.23–0.33 | 0.708 | 1386.28 | −0.01 | −0.16–0.13 | 0.846 | 1388.84 | −0.04 | −0.29–0.21 | 0.754 | 1386.02 |
| dummy(School level)6 | −0.13 | −0.35–0.08 | 0.221 | 1386.47 | 0.02 | −0.25–0.30 | 0.871 | 1386.36 | −0.01 | −0.14–0.13 | 0.936 | 1388.97 | −0.04 | −0.28–0.20 | 0.751 | 1386.02 |
| dummy(School level)7 | −0.14 | −0.35–0.07 | 0.193 | 1386.45 | 0.05 | −0.23–0.32 | 0.745 | 1386.34 | −0.01 | −0.15–0.13 | 0.898 | 1388.96 | −0.04 | −0.28–0.20 | 0.754 | 1386.02 |
| dummy(School level)8 | −0.14 | −0.36–0.07 | 0.181 | 1386.38 | 0.03 | −0.24–0.31 | 0.809 | 1386.29 | −0.01 | −0.14–0.13 | 0.928 | 1388.97 | −0.05 | −0.29–0.20 | 0.713 | 1386.02 |
| dummy(School level)9 | −0.21 | −0.51–0.09 | 0.163 | 1386.59 | −0.09 | −0.47–0.30 | 0.659 | 1386.45 | −0.1 | −0.30–0.09 | 0.29 | 1387.6 | −0.05 | −0.40–0.29 | 0.754 | 1386.03 |
| Childhood adversity [1] | −0.02 | −0.04 to −0.00 | 0.017 | 1386.81 | −0.02 | −0.04 to −0.00 | 0.024 | 1386.55 | −0.01 | −0.02–0.00 | 0.178 | 1386.58 | −0.02 | −0.04 to −0.00 | 0.024 | 1386.03 |

**Random effects**

| | | | | | | | | | | | | | | | | |
|---|---|---|---|---|---|---|---|---|---|---|---|---|---|---|---|---|
| $\sigma^2$ | 0.01 | | | | 0.02 | | | | 0 | | | | 0.02 | | | |
| $\tau_{00}$ | 0.00 Site | | | | 0.00 Site | | | | 0.00 Site | | | | 0.01 Site | | | |
| ICC | 0.03 | | | | 0.04 | | | | | | | | 0.41 | | | |
| N | 4 Site | | | | 4 Site | | | | 4 Site | | | | 4 Site | | | |
| Observations | 1402 | | | | 1402 | | | | 1402 | | | | 1402 | | | |
| Marginal R²/Conditional R² | 0.021/0.049 | | | | 0.012/0.049 | | | | 0.006/NA | | | | 0.004/0.414 | | | |

**B**

| Predictors | CS discrimination arousal ratings during ACQ | | | | CS discrimination arousal ratings during GEN | | | | LDS arousal ratings | | | | General reactivity arousal ratings | | | |
|---|---|---|---|---|---|---|---|---|---|---|---|---|---|---|---|---|
| | Estimates | CI | p | df | Estimates | CI | p | df | Estimates | CI | p | df | Estimates | CI | p | df |
| (Intercept) | 4.73 | 0.46–9.00 | 0.03 | 1387.12 | 3.89 | −0.92–8.69 | 0.113 | 1386.01 | 1.37 | −1.27–4.01 | 0.308 | 1388.45 | 3.22 | 0.85–5.59 | 0.008 | 1333.36 |
| Age | −0.02 | −0.04–0.01 | 0.19 | 1381.61 | 0 | −0.02–0.03 | 0.854 | 1384.27 | 0.01 | −0.00–0.03 | 0.063 | 1213.66 | −0.01 | −0.03 to −0.00 | 0.034 | 1388.87 |
| Sex [1] | −0.72 | −0.96 to −0.49 | <0.001 | 1380.21 | −0.68 | −0.94 to −0.42 | <0.001 | 1383.34 | −0.1 | −0.24–0.05 | 0.198 | 1151.26 | −0.22 | −0.35 to −0.09 | 0.001 | 1388.9 |
| dummy(School level)1 | −2.02 | −7.98–3.95 | 0.507 | 1386.01 | 1 | −5.70–7.70 | 0.769 | 1386.01 | −1.49 | −5.17–2.19 | 0.428 | 1386.12 | 1.74 | −1.56–5.03 | 0.301 | 1386 |
| dummy(School level)2 | −1.93 | −6.33–2.48 | 0.391 | 1387.53 | −1.08 | −6.03–3.87 | 0.668 | 1387.35 | −1.2 | −3.92–1.52 | 0.388 | 1388.08 | 0.66 | −1.78–3.09 | 0.596 | 1386.48 |
| dummy(School level)3 | −1.96 | −6.58–2.67 | 0.407 | 1387.23 | −1.29 | −6.49–3.91 | 0.627 | 1387.07 | −1.02 | −3.88–1.84 | 0.484 | 1388.8 | 0.45 | −2.11–3.01 | 0.73 | 1386.37 |
| dummy(School level)4 | −1.46 | −5.71–2.79 | 0.499 | 1387.45 | −0.86 | −5.64–3.91 | 0.723 | 1387.28 | −1.2 | −3.82–1.43 | 0.372 | 1388.6 | 1.26 | −1.09–3.61 | 0.294 | 1386.46 |
| dummy(School level)5 | −0.71 | −5.06–3.64 | 0.747 | 1386.81 | −0.42 | −5.31–4.47 | 0.868 | 1386.71 | −1.8 | −4.49 to −0.89 | 0.189 | 1388.62 | 1.57 | −0.83–3.98 | 0.2 | 1386.25 |

*Appendix 1—table 2 continued on next page*

*Appendix 1—table 2 continued*

**B**

| Predictors | CS discrimination arousal ratings during ACQ | | | | CS discrimination arousal ratings during GEN | | | | LDS arousal ratings | | | | General reactivity arousal ratings | | | |
|---|---|---|---|---|---|---|---|---|---|---|---|---|---|---|---|---|
| | Estimates | CI | p | df | Estimates | CI | p | df | Estimates | CI | p | df | Estimates | CI | p | df |
| dummy(School level)6 | −0.8 | −5.06–3.46 | 0.712 | 1387.04 | −0.48 | −5.26–4.31 | 0.845 | 1386.91 | −1.34 | −3.97–1.29 | 0.317 | 1388.98 | 1.16 | −1.19–3.51 | 0.333 | 1386.32 |
| dummy(School level)7 | −1.27 | −5.49–2.96 | 0.556 | 1387.01 | −0.47 | −5.22–4.27 | 0.845 | 1386.89 | −1.19 | −3.80–1.42 | 0.373 | 1388.98 | 1.28 | −1.05–3.62 | 0.281 | 1386.3 |
| dummy(School level)8 | −1.29 | −5.52–2.93 | 0.549 | 1386.85 | −0.42 | −5.17–4.33 | 0.862 | 1386.75 | −1.23 | −3.84–1.38 | 0.356 | 1388.8 | 1.29 | −1.04–3.63 | 0.278 | 1386.26 |
| dummy(School level)9 | 1.99 | −3.99–7.96 | 0.514 | 1387.32 | 2.92 | −3.79–9.64 | 0.393 | 1387.16 | −3.15 | −6.84–0.55 | 0.095 | 1388.59 | 1.59 | −1.72–4.89 | 0.346 | 1386.4 |
| Childhood adversity [1] | 0.26 | −0.06–0.59 | 0.108 | 1387.55 | 0.06 | −0.30–0.42 | 0.752 | 1387.37 | 0.04 | −0.16–0.24 | 0.67 | 1388.05 | 0.03 | −0.15–0.21 | 0.738 | 1386.48 |
| **Random effects** | | | | | | | | | | | | | | | | |
| σ² | 4.62 | | | | 5.83 | | | | 1.76 | | | | 1.41 | | | |
| τ00 | 0.05 Site | | | | 0.08 Site | | | | 0.00 Site | | | | 0.06 Site | | | |
| ICC | 0.01 | | | | 0.01 | | | | 0 | | | | 0.04 | | | |
| N | 4 Site | | | | 4 Site | | | | 4 Site | | | | 4 Site | | | |
| Observations | 1402 | | | | 1402 | | | | 1402 | | | | 1402 | | | |
| Marginal R²/Conditional R² | 0.036/0.046 | | | | 0.021/0.034 | | | | 0.009/0.010 | | | | 0.019/0.061 | | | |

**C**

| Predictors | CS discrimination valence ratings during ACQ | | | | CS discrimination valence ratings during GEN | | | | LDS valence ratings | | | | General reactivity valence ratings | | | |
|---|---|---|---|---|---|---|---|---|---|---|---|---|---|---|---|---|
| | Estimates | CI | p | df | Estimates | CI | p | df | Estimates | CI | p | df | Estimates | CI | p | df |
| (Intercept) | 3.58 | −0.74–7.90 | 0.104 | 1377.81 | 3.4 | −1.67–8.48 | 0.189 | 1388.66 | 2.39 | −0.25–5.02 | 0.076 | 1382.58 | 5.82 | 3.94–7.70 | <0.001 | 1365.32 |
| Age | −0.03 | −0.06 to −0.01 | 0.005 | 1388.31 | −0.02 | −0.04–0.01 | 0.266 | 1243.67 | 0 | −0.01–0.02 | 0.884 | 1387.16 | −0.01 | −0.02–0.00 | 0.062 | 1388.96 |
| Sex [1] | −0.55 | −0.79 to −0.31 | <0.001 | 1388.11 | −0.66 | −0.94 to −0.38 | <0.001 | 1195.8 | −0.07 | −0.21–0.08 | 0.352 | 1386.74 | −0.01 | −0.11–0.10 | 0.916 | 1388.92 |
| dummy(School level)1 | −2.03 | −8.05–3.98 | 0.507 | 1386 | −1.02 | −8.11–6.08 | 0.779 | 1386.1 | −1.75 | −5.42–1.93 | 0.351 | 1386 | −0.84 | −3.46–1.77 | 0.527 | 1386 |
| dummy(School level)2 | −2.53 | −6.98–1.91 | 0.263 | 1386.89 | −1.19 | −6.44–4.05 | 0.655 | 1388.63 | −2.08 | −4.79–0.64 | 0.134 | 1387.08 | −1.14 | −3.07–0.79 | 0.246 | 1386.68 |
| dummy(School level)3 | −0.48 | −5.15–4.19 | 0.84 | 1386.7 | −1.2 | −6.71–4.30 | 0.668 | 1388.97 | −1.33 | −4.18–1.52 | 0.361 | 1386.84 | −1.55 | −3.58–0.48 | 0.134 | 1386.52 |
| dummy(School level)4 | −0.31 | −4.59–3.98 | 0.889 | 1386.85 | 0.19 | −4.87–5.25 | 0.941 | 1388.88 | −1.91 | −4.53–0.71 | 0.154 | 1387.02 | −0.85 | −2.72–1.01 | 0.371 | 1386.65 |
| dummy(School level)5 | 0.07 | −4.32–4.46 | 0.976 | 1386.46 | 0.58 | −4.60–5.75 | 0.827 | 1388.45 | −2.3 | −4.99–0.38 | 0.092 | 1386.56 | −0.36 | −2.27–1.55 | 0.711 | 1386.35 |
| dummy(School level)6 | 0.07 | −4.22–4.37 | 0.973 | 1386.6 | 0.11 | −4.95–5.18 | 0.965 | 1388.89 | −1.81 | −4.44–0.81 | 0.176 | 1386.72 | −0.91 | −2.78–0.96 | 0.338 | 1386.45 |
| dummy(School level)7 | −0.53 | −4.79–3.73 | 0.807 | 1386.57 | −0.12 | −5.14–4.91 | 0.964 | 1388.89 | −1.85 | −4.46–0.75 | 0.163 | 1386.7 | −0.74 | −2.59–1.11 | 0.433 | 1386.43 |
| dummy(School level)8 | −0.4 | −4.67–3.86 | 0.853 | 1386.48 | 0.04 | −4.99–5.07 | 0.987 | 1388.64 | −1.83 | −4.43–0.78 | 0.17 | 1386.59 | −0.75 | −2.61–1.10 | 0.426 | 1386.36 |
| dummy(School level)9 | 2.2 | −3.83–8.23 | 0.474 | 1386.75 | 2.68 | −4.43–9.80 | 0.459 | 1388.9 | −3.61 | −7.30–0.07 | 0.055 | 1386.91 | −0.41 | −3.04–2.21 | 0.757 | 1386.57 |
| Childhood adversity [1] | −0.02 | −0.35–0.31 | 0.902 | 1386.9 | −0.01 | −0.40–0.37 | 0.95 | 1388.6 | −0.03 | −0.23–0.17 | 0.754 | 1387.09 | −0.04 | −0.18–0.10 | 0.581 | 1386.69 |
| **Random effects** | | | | | | | | | | | | | | | | |
| σ² | 4.7 | | | | 6.53 | | | | 1.76 | | | | 0.89 | | | |
| τ00 | 0.10 Site | | | | 0.01 Site | | | | 0.03 Site | | | | 0.03 Site | | | |
| ICC | 0.02 | | | | 0 | | | | 0.02 | | | | 0.03 | | | |

*Appendix 1—table 2 continued on next page*

*Appendix 1—table 2 continued*

**C**

| Predictors | CS discrimination valence ratings during ACQ | | | | CS discrimination valence ratings during GEN | | | | LDS valence ratings | | | | General reactivity valence ratings | | | |
|---|---|---|---|---|---|---|---|---|---|---|---|---|---|---|---|---|
| | Estimates | CI | p | df | Estimates | CI | p | df | Estimates | CI | p | df | Estimates | CI | p | df |
| N | 4 Site | | | | 4 Site | | | | 4 Site | | | | 4 Site | | | |
| Observations | 1402 | | | | 1402 | | | | 1402 | | | | 1402 | | | |
| Marginal R²/Conditional R² | 0.035/0.055 | | | | 0.021/0.022 | | | | 0.006/0.023 | | | | 0.015/0.044 | | | |

**D**

| Predictors | CS discrimination contingency ratings during ACQ | | | | CS discrimination contingency ratings during GEN | | | | LDS contingency ratings | | | | General reactivity contingency ratings | | | |
|---|---|---|---|---|---|---|---|---|---|---|---|---|---|---|---|---|
| | Estimates | CI | p | df | Estimates | CI | p | df | Estimates | CI | p | df | Estimates | CI | p | df |
| (Intercept) | 5.56 | −57.39–68.52 | 0.862 | 1381.99 | 83.3 | 25.26–141.34 | **0.005** | 1384.67 | 43.25 | 10.35–76.14 | **0.01** | 1379.65 | 21.72 | −3.20–46.65 | 0.088 | 1169.04 |
| Age | −0.18 | −0.53–0.16 | 0.303 | 1387.39 | 0.2 | −0.12–0.52 | 0.217 | 1385.88 | 0.03 | −0.15–0.21 | 0.728 | 1388.01 | 0 | −0.15–0.15 | 0.963 | 1170.37 |
| Sex [1] | −6.28 | −9.74 to −2.82 | **<0.001** | 1387.01 | −3.72 | −6.91 to −0.53 | **0.022** | 1385.23 | 0.58 | −1.22–2.39 | 0.526 | 1387.75 | −1.58 | −3.05 to −0.11 | **0.036** | 1159.35 |
| dummy(School level)1 | 94.82 | 7.05–182.59 | **0.034** | 1386 | 0.2 | −80.74–81.14 | 0.996 | 1386.01 | 0.03 | −45.82–45.88 | 0.999 | 1386 | 9.17 | −25.54–43.88 | 0.604 | 1169 |
| dummy(School level)2 | 21.75 | −43.07–86.58 | 0.51 | 1387.05 | −45.11 | −104.90–14.67 | 0.139 | 1387.21 | −30.03 | −63.89–3.83 | 0.082 | 1386.95 | 9.97 | −15.78–35.73 | 0.447 | 1170.68 |
| dummy(School level)3 | 35.78 | −32.32–103.88 | 0.303 | 1386.82 | −34.55 | −97.35–28.25 | 0.281 | 1386.96 | −29.19 | −64.76–6.38 | 0.108 | 1386.74 | 13.87 | −13.07–40.81 | 0.313 | 1170.47 |
| dummy(School level)4 | 46.02 | −16.56–108.60 | 0.149 | 1386.99 | −32.68 | −90.40–25.03 | 0.267 | 1387.15 | −29.54 | −62.23–3.15 | 0.076 | 1386.9 | 20.52 | −4.28–45.31 | 0.105 | 1170.59 |
| dummy(School level)5 | 55.51 | −8.54–119.57 | 0.089 | 1386.54 | −35.9 | −94.97–23.17 | 0.233 | 1386.63 | −35.79 | −69.25 to −2.33 | **0.036** | 1386.49 | 23.91 | −1.66–49.49 | 0.067 | 1169.58 |
| dummy(School level)6 | 49.68 | −12.99–112.35 | 0.12 | 1386.7 | −31.66 | −89.46–26.13 | 0.283 | 1386.82 | −32.52 | −65.26–0.22 | 0.052 | 1386.64 | 21.1 | −3.75–45.94 | 0.096 | 1170.1 |
| dummy(School level)7 | 53.68 | −8.52–115.88 | 0.091 | 1386.68 | −29.33 | −86.69–28.03 | 0.316 | 1386.79 | −30.67 | −63.16–1.82 | 0.064 | 1386.61 | 18.22 | −6.39–42.83 | 0.147 | 1170.18 |
| dummy(School level)8 | 56.22 | −6.02–118.46 | 0.077 | 1386.57 | −30.69 | −88.08–26.71 | 0.294 | 1386.67 | −30.51 | −63.02–2.00 | 0.066 | 1386.51 | 17.44 | −7.18–42.07 | 0.165 | 1169.91 |
| dummy(School level)9 | 89.88 | 1.91–177.85 | **0.045** | 1386.89 | −57.56 | −138.69–23.57 | 0.164 | 1387.04 | −35.92 | −81.87–10.03 | 0.125 | 1386.8 | −0.23 | −35.05–34.58 | 0.989 | 1170.57 |
| Childhood adversity [1] | −0.86 | −5.61–3.89 | 0.723 | 1387.06 | −1.26 | −5.65–3.12 | 0.572 | 1387.23 | 1.13 | −1.35–3.62 | 0.37 | 1386.96 | 0.77 | −1.27–2.81 | 0.461 | 1170.8 |
| **Random effects** | | | | | | | | | | | | | | | | |
| σ² | 1001 | | | | 851.3 | | | | 273.12 | | | | 156.5 | | | |
| τ₀₀ | 18.22 Site | | | | 12.82 Site | | | | 5.58 Site | | | | 0.88 Site | | | |
| ICC | 0.02 | | | | 0.01 | | | | 0.02 | | | | 0.01 | | | |
| N | 4 Site | | | | 4 Site | | | | 4 Site | | | | 3 Site | | | |
| Observations | 1402 | | | | 1402 | | | | 1402 | | | | 1184 | | | |
| Marginal R²/Conditional R² | 0.029/0.047 | | | | 0.009/0.024 | | | | 0.008/0.028 | | | | 0.020/0.025 | | | |

Note. Due to its categorical nature, we included school level as a dummy variable.

**Appendix 1—table 3.** Exploratory results of testing the cumulative risk model involving severity groups using ANOVAs with exposure to abuse as between-subject factor and conditioned stimulus (CS) discrimination, LDS, and general reactivity as dependent variable.

| Outcome | Phase | Measure | Mean 'none' | SD 'none' | Mean 'low' | SD 'low' | Mean 'moderate' | SD 'moderate' | Mean 'severe' | SD 'severe' | $df_{Num}$ | $df_{Den}$ | F | p | partial Eta² |
|---|---|---|---|---|---|---|---|---|---|---|---|---|---|---|---|
| CS discrimination | ACQ | SCR | 0.05 | 0.11 | 0.04 | 0.10 | 0.02 | 0.11 | 0.03 | 0.10 | 3 | 1,398 | 2.35 | 0.071 | 0.00 |
| | | Arousal ratings | 2.80 | 2.21 | 2.76 | 2.18 | 2.99 | 2.07 | 3.13 | 2.21 | 3 | 1,398 | 0.86 | 0.459 | 0.00 |
| | | Valence ratings | 2.05 | 2.19 | 2.21 | 2.21 | 2.11 | 2.21 | 2.00 | 2.41 | 3 | 1,398 | 0.54 | 0.655 | 0.00 |
| | | Contingency ratings | 53.01 | 31.88 | 52.06 | 32.00 | 52.37 | 33.16 | 48.03 | 35.02 | 3 | 1,398 | 0.51 | 0.673 | 0.00 |
| | GEN | SCR | 0.05 | 0.14 | 0.05 | 0.14 | 0.01 | 0.12 | 0.04 | 0.14 | 3 | 1,398 | 2.36 | 0.070 | 0.00 |
| | | Arousal ratings | 3.23 | 2.44 | 3.02 | 2.42 | 3.12 | 2.35 | 3.39 | 2.77 | 3 | 1,398 | 0.91 | 0.437 | 0.00 |
| | | Valence ratings | 2.66 | 2.64 | 2.73 | 2.41 | 2.66 | 2.53 | 2.71 | 2.89 | 3 | 1,398 | 0.07 | 0.975 | 0.00 |
| | | Contingency ratings | 58.20 | 28.95 | 56.12 | 28.66 | 56.93 | 30.43 | 54.55 | 35.09 | 3 | 137 | 0.80 | 0.498 | 0.13 |
| LDS | GEN | SCR | 0.01 | 0.07 | 0.01 | 0.07 | 0.00 | 0.06 | 0.00 | 0.06 | 3 | 1,398 | 0.60 | 0.615 | 0.00 |
| | | Arousal ratings | 0.52 | 1.32 | 0.38 | 1.32 | 0.48 | 1.40 | 0.63 | 1.25 | 3 | 1,398 | 1.38 | 0.248 | 0.00 |
| | | Valence ratings | 0.56 | 1.33 | 0.54 | 1.33 | 0.50 | 1.23 | 0.59 | 1.49 | 3 | 1,398 | 0.14 | 0.935 | 0.00 |
| | | Contingency ratings | 14.11 | 16.42 | 13.91 | 16.79 | 14.82 | 17.43 | 16.10 | 16.32 | 3 | 1,398 | 0.40 | 0.754 | 0.00 |
| General reactivity | ALL | SCR | 0.10 | 0.16 | 0.09 | 0.12 | 0.07 | 0.10 | 0.07 | 0.10 | 3 | 1,398 | 1.64 | 0.178 | 0.00 |
| | | Arousal ratings | 4.01 | 1.22 | 4.04 | 1.24 | 4.15 | 1.09 | 3.79 | 1.12 | 3 | 1,398 | 1.43 | 0.234 | 0.00 |
| | | Valence ratings | 4.81 | 0.96 | 4.77 | 0.98 | 4.78 | 0.83 | 4.64 | 0.92 | 3 | 1,398 | 0.78 | 0.508 | 0.00 |
| | | Contingency ratings | 39.42 | 12.21 | 39.09 | 13.71 | 40.54 | 11.64 | 39.50 | 11.32 | 3 | 120 | 0.54 | 0.656 | 0.15 |

Note. ACQ = acquisition training, GEN = generalization phase, LDS = linear deviation score, SD = standard deviation. Italic lines indicate the application of robust ANOVAs. In this context, effect sizes do not indicate partial eta squared, but the explanatory measure of effect size (**Mair and Wilcox, 2020**). Values of 0.10, 0.30, and 0.50 represent small, medium, and large effect sizes, respectively.

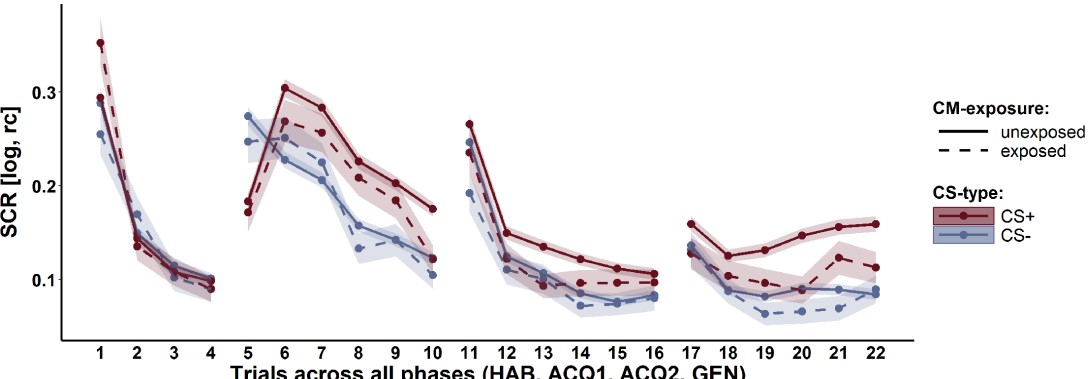

**Appendix 1—figure 4.** Trial-by-trial skin conductance response (SCR) data across all experimental phases for the CS+ (red) and CS- (blue) for individuals exposed (dashed lines) and unexposed (solid lines) to childhood adversity separately. Ribbons represent standard errors of the means (SEMs) including $n_{unexposed}$ = 1199 and $n_{exposed}$ = 203.

**Appendix 1—table 4.** Exploratory results of testing the cumulative risk model involving the number of subscales exceeding an at least moderate cut-off using regressions with the number of subscales as predictor and conditioned stimulus (CS) discrimination, LDS, and general reactivity as criterion.

| Outcome | Phase | Measure | beta | $SE_b$ | LL (95% CI) | UL (95% CI) | Beta | t | df | p | $R^2$ | Cohen's $f^2$ |
|---|---|---|---|---|---|---|---|---|---|---|---|---|
| CS discrimination | ACQ | SCR | –0.01 | 0.00 | –0.02 | 0.00 | –0.07 | –2.62 | 1,400 | **0.009** | 0 | 0 |
| | | Arousal ratings | 0.07 | 0.09 | –0.09 | 0.24 | 0.02 | 0.85 | 1,400 | 0.393 | 0 | 0 |
| | | Valence ratings | –0.05 | 0.09 | –0.22 | 0.12 | –0.01 | –0.55 | 1,400 | 0.582 | 0 | 0 |
| | | Contingency ratings | –2.53 | 1.25 | –4.98 | –0.07 | –0.05 | –2.02 | 1,400 | **0.044** | 0 | 0 |
| | GEN | SCR | –0.01 | 0.00 | –0.02 | 0.00 | –0.03 | –1.23 | 1,400 | 0.218 | 0 | 0 |
| | | Arousal ratings | 0.00 | 0.10 | –0.19 | 0.19 | 0.00 | 0.00 | 1,400 | 0.997 | 0 | 0 |
| | | Valence ratings | 0.03 | 0.10 | –0.17 | 0.22 | 0.01 | 0.27 | 1,400 | 0.789 | 0 | 0 |
| | | Contingency ratings | –1.61 | 1.14 | –3.85 | 0.63 | –0.04 | –1.41 | 1,400 | 0.159 | 0 | 0 |
| LDS | GEN | SCR | 0.00 | 0.00 | –0.01 | 0.00 | –0.01 | –0.48 | 1,400 | 0.629 | 0 | 0 |
| | | Arousal ratings | 0.03 | 0.05 | –0.07 | 0.13 | 0.01 | 0.56 | 1,400 | 0.579 | 0 | 0 |
| | | Valence ratings | 0.00 | 0.05 | –0.10 | 0.10 | 0.00 | 0.04 | 1,400 | 0.965 | 0 | 0 |
| | | Contingency ratings | 0.71 | 0.65 | –0.56 | 1.98 | 0.03 | 1.10 | 1,400 | 0.272 | 0 | 0 |
| General reactivity | ALL | SCR | –0.01 | 0.00 | –0.02 | 0.00 | –0.04 | –1.91 | 1,400 | 0.057 | 0 | 0 |
| | | Arousal ratings | –0.03 | 0.05 | –0.12 | 0.06 | –0.02 | –0.65 | 1,400 | 0.517 | 0 | 0 |
| | | Valence ratings | –0.04 | 0.04 | –0.11 | 0.03 | –0.03 | –1.06 | 1,400 | 0.290 | 0 | 0 |
| | | Contingency ratings | 0.59 | 0.55 | –0.48 | 1.66 | 0.03 | 1.08 | 1,182 | 0.281 | 0 | 0 |

Note. ACQ = acquisition training, GEN = generalization phase, LDS = linear deviation score. Bold numbers indicate significant results (p<0.05).

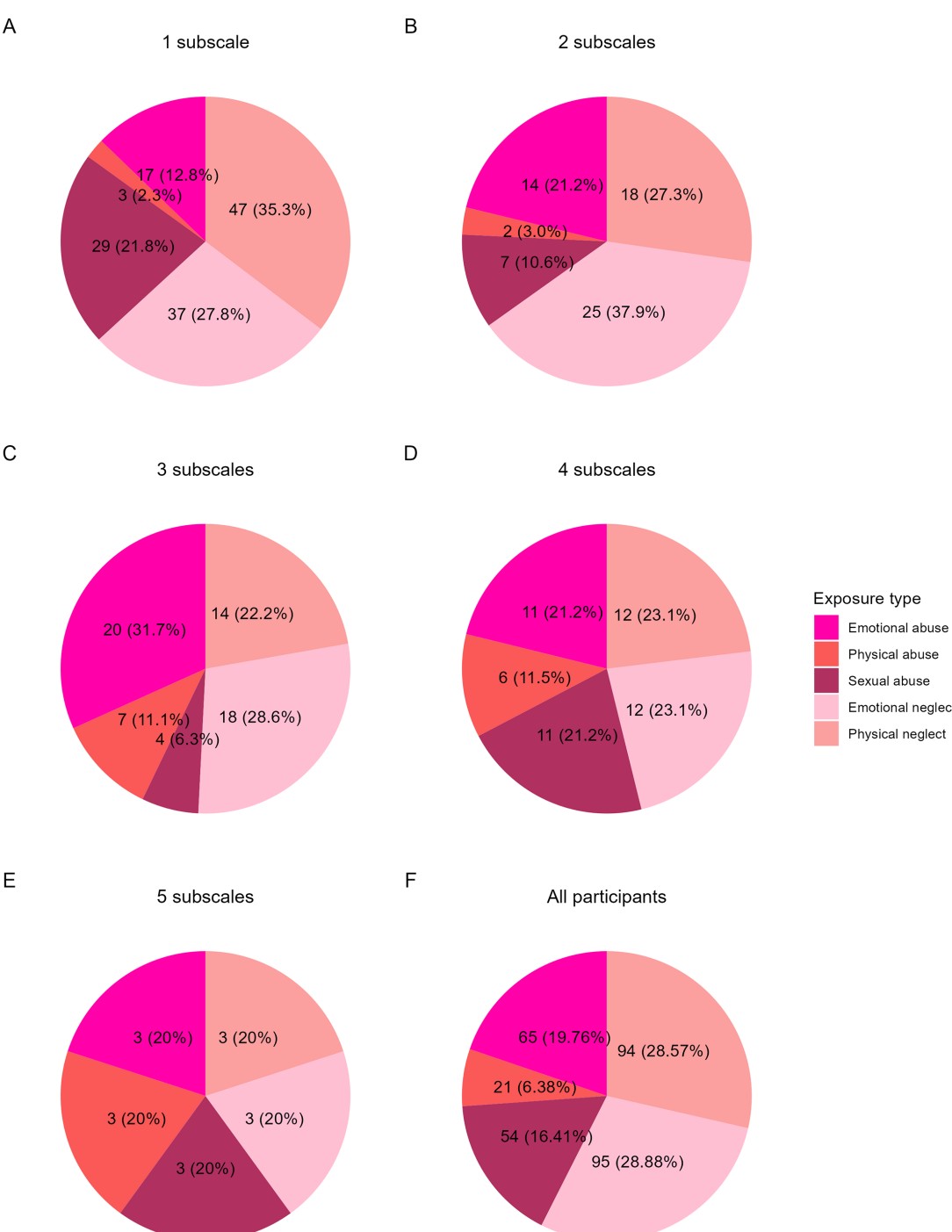

**Appendix 1—figure 5.** Illustration of the distribution of the different numbers (i.e. 1–5) of exceeded subscales among the Childhood Trauma Questionnaire (CTQ) exposure types emotional abuse, physical abuse, emotional neglect, and physical neglect (**A-E**) and distribution of exposure types across all participants (**F**).

**Appendix 1—table 5.** Exploratory results of testing the specificity model using regressions with exposure to abuse as predictor and conditioned stimulus (CS) discrimination, LDS, and general reactivity as criterion.

| Outcome | Phase | Measure | beta | SE$_b$ | LL (95% CI) | UL (95% CI) | Beta | t | df | p | R² | Cohen's f² |
|---|---|---|---|---|---|---|---|---|---|---|---|---|
| CS discrimination | ACQ | SCR | 0.00 | 0.00 | 0.00 | 0.00 | –0.03 | –1.26 | 1,400 | 0.209 | 0 | 0 |
| | | Arousal ratings | 0.01 | 0.01 | –0.02 | 0.03 | 0.02 | 0.59 | 1,400 | 0.556 | 0 | 0 |
| | | Valence ratings | 0.00 | 0.01 | –0.02 | 0.03 | 0.01 | 0.27 | 1,400 | 0.789 | 0 | 0 |
| | | Contingency ratings | –0.25 | 0.20 | –0.64 | 0.14 | –0.03 | –1.25 | 1,400 | 0.211 | 0 | 0 |
| | GEN | SCR | 0.00 | 0.00 | 0.00 | 0.00 | 0.03 | 1.01 | 1,400 | 0.311 | 0 | 0 |
| | | Arousal ratings | 0.00 | 0.01 | –0.02 | 0.04 | 0.01 | 0.35 | 1,400 | 0.725 | 0 | 0 |
| | | Valence ratings | 0.02 | 0.02 | –0.01 | 0.05 | 0.03 | 1.20 | 1,400 | 0.230 | 0 | 0 |
| | | Contingency ratings | –0.13 | 0.18 | –0.48 | 0.23 | –0.02 | –0.71 | 1,400 | 0.479 | 0 | 0 |
| LDS | GEN | SCR | 0.00 | 0.00 | 0.00 | 0.00 | –0.01 | –0.22 | 1,400 | 0.825 | 0 | 0 |
| | | Arousal ratings | 0.00 | 0.01 | –0.02 | 0.01 | 0.00 | –0.08 | 1,400 | 0.933 | 0 | 0 |
| | | Valence ratings | 0.00 | 0.01 | –0.02 | 0.01 | –0.01 | –0.47 | 1,400 | 0.639 | 0 | 0 |
| | | Contingency ratings | 0.07 | 0.10 | –0.13 | 0.27 | 0.02 | 0.68 | 1,400 | 0.496 | 0 | 0 |
| General reactivity | ALL | SCR | 0.00 | 0.00 | 0.00 | 0.00 | 0.00 | –0.02 | 1,400 | 0.984 | 0 | 0 |
| | | Arousal ratings | 0.00 | 0.01 | –0.01 | 0.02 | 0.02 | 0.66 | 1,400 | 0.511 | 0 | 0 |
| | | Valence ratings | 0.00 | 0.01 | –0.01 | 0.01 | 0.00 | –0.19 | 1,400 | 0.853 | 0 | 0 |
| | | Contingency ratings | 0.12 | 0.09 | –0.05 | 0.29 | 0.04 | 1.38 | 1,182 | 0.167 | 0 | 0 |

Note. ACQ = acquisition training, GEN = generalization phase, LDS = linear deviation score.

**Appendix 1—table 6.** Exploratory results of testing the specificity model using regressions with exposure to neglect as predictor and conditioned stimulus (CS) discrimination, LDS, and general reactivity as criterion.

| Outcome | Phase | Measure | beta | SE$_b$ | LL (95% CI) | UL (95% CI) | Beta | t | df | p | R² | Cohen's f² |
|---|---|---|---|---|---|---|---|---|---|---|---|---|
| CS discrimination | ACQ | SCR | **0.00** | **0.00** | **0.00** | **0.00** | **–0.07** | **–2.53** | **1,400** | **0.012** | 0 | 0 |
| | | Arousal ratings | 0.01 | 0.01 | –0.02 | 0.03 | 0.01 | 0.50 | 1,400 | 0.615 | 0 | 0 |
| | | Valence ratings | 0.00 | 0.01 | –0.02 | 0.03 | 0.00 | 0.10 | 1,400 | 0.919 | 0 | 0 |
| | | Contingency ratings | **–0.41** | **0.17** | **–0.75** | **–0.07** | **–0.06** | **–2.36** | **1,400** | **0.018** | 0 | 0 |
| | GEN | SCR | 0.00 | 0.00 | 0.00 | 0.00 | –0.04 | –1.29 | 1,400 | 0.196 | 0 | 0 |
| | | Arousal ratings | –0.01 | 0.01 | –0.03 | 0.02 | –0.02 | –0.59 | 1,400 | 0.558 | 0 | 0 |
| | | Valence ratings | 0.00 | 0.01 | –0.03 | 0.02 | –0.01 | –0.27 | 1,400 | 0.789 | 0 | 0 |
| | | Contingency ratings | **–0.40** | **0.16** | **–0.71** | **–0.09** | **–0.07** | **–2.52** | **1,400** | **0.012** | 0 | 0 |
| LDS | GEN | SCR | 0.00 | 0.00 | 0.00 | 0.00 | –0.02 | –0.66 | 1,400 | 0.512 | 0 | 0 |
| | | Arousal ratings | 0.00 | 0.01 | –0.02 | 0.01 | –0.01 | –0.25 | 1,400 | 0.799 | 0 | 0 |
| | | Valence ratings | 0.00 | 0.01 | –0.01 | 0.01 | 0.00 | 0.08 | 1,400 | 0.935 | 0 | 0 |
| | | Contingency ratings | 0.01 | 0.09 | –0.17 | 0.19 | 0.00 | 0.11 | 1,400 | 0.915 | 0 | 0 |
| General reactivity | ALL | SCR | **0.00** | **0.00** | **0.00** | **0.00** | **–0.06** | **–2.31** | **1,400** | **0.021** | 0 | 0 |
| | | Arousal ratings | 0.00 | 0.01 | –0.02 | 0.01 | –0.02 | –0.67 | 1,400 | 0.504 | 0 | 0 |
| | | Valence ratings | –0.01 | 0.00 | –0.02 | 0.00 | –0.05 | –1.76 | 1,400 | 0.079 | 0 | 0 |
| | | Contingency ratings | 0.07 | 0.07 | –0.08 | 0.22 | 0.03 | 0.91 | 1,182 | 0.365 | 0 | 0 |

Note. ACQ = acquisition training, GEN = generalization phase, LDS = linear deviation score. Bold numbers indicate significant results (p<0.05).

**Appendix 1—table 7.** Exploratory results of testing the dimensional model using multiple regressions with exposure to both abuse and neglect as predictors and conditioned stimulus (CS) discrimination, LDS, and general reactivity as criterion.

| Outcome | Phase | Measure | predictor | beta | SE$_b$ | LL (95% CI) | UL (95% CI) | Beta | t | df | p | adj. R² | Cohen's f² |
|---|---|---|---|---|---|---|---|---|---|---|---|---|---|
| CS discrimination | ACQ | SCR | abuse | 0.00 | 0.00 | −0.01 | 0.00 | −0.09 | −1.19 | 1,398 | 0.234 | 0 | 0 |
| | | | neglect | 0.00 | 0.00 | −0.01 | 0.00 | −0.17 | −2.33 | 1,398 | 0.020 | 0 | 0 |
| | | | interaction | 0.00 | 0.00 | 0.00 | 0.00 | 0.19 | 1.49 | 1,398 | 0.137 | 0 | 0 |
| | | Arousal ratings | abuse | 0.05 | 0.04 | −0.03 | 0.12 | 0.09 | 1.17 | 1,398 | 0.240 | 0 | 0 |
| | | | neglect | 0.04 | 0.03 | −0.03 | 0.10 | 0.08 | 1.09 | 1,398 | 0.275 | 0 | 0 |
| | | | interaction | 0.00 | 0.00 | 0.00 | 0.00 | −0.14 | −1.13 | 1,398 | 0.258 | 0 | 0 |
| | | Valence ratings | abuse | 0.04 | 0.04 | −0.04 | 0.12 | 0.08 | 1.03 | 1,398 | 0.303 | 0 | 0 |
| | | | neglect | 0.03 | 0.03 | −0.04 | 0.10 | 0.07 | 0.88 | 1,398 | 0.382 | 0 | 0 |
| | | | interaction | 0.00 | 0.00 | 0.00 | 0.00 | −0.13 | −1.02 | 1,398 | 0.309 | 0 | 0 |
| | | Contingency ratings | abuse | 0.81 | 0.58 | −0.33 | 1.95 | 0.11 | 1.39 | 1,398 | 0.165 | 0 | 0 |
| | | | neglect | 0.18 | 0.49 | −0.78 | 1.14 | 0.03 | 0.37 | 1,398 | 0.712 | 0 | 0 |
| | | | interaction | −0.03 | 0.02 | −0.08 | 0.01 | −0.18 | −1.43 | 1,398 | 0.154 | 0 | 0 |
| | GEN | SCR | abuse | 0.00 | 0.00 | 0.00 | 0.01 | 0.05 | 0.64 | 1,398 | 0.525 | 0 | 0 |
| | | | neglect | 0.00 | 0.00 | −0.01 | 0.00 | −0.11 | −1.42 | 1,398 | 0.156 | 0 | 0 |
| | | | interaction | 0.00 | 0.00 | 0.00 | 0.00 | 0.05 | 0.38 | 1,398 | 0.705 | 0 | 0 |
| | | Arousal ratings | abuse | 0.03 | 0.04 | −0.06 | 0.12 | 0.05 | 0.63 | 1,398 | 0.527 | 0 | 0 |
| | | | neglect | −0.01 | 0.04 | −0.08 | 0.06 | −0.01 | −0.20 | 1,398 | 0.838 | 0 | 0 |
| | | | interaction | 0.00 | 0.00 | 0.00 | 0.00 | −0.04 | −0.28 | 1,398 | 0.782 | 0 | 0 |
| | | Valence ratings | abuse | 0.04 | 0.05 | −0.05 | 0.13 | 0.06 | 0.83 | 1,398 | 0.405 | 0 | 0 |
| | | | neglect | −0.02 | 0.04 | −0.10 | 0.06 | −0.04 | −0.47 | 1,398 | 0.636 | 0 | 0 |
| | | | interaction | 0.00 | 0.00 | 0.00 | 0.00 | −0.01 | −0.10 | 1,398 | 0.918 | 0 | 0 |
| | | Contingency ratings | abuse | 0.56 | 0.53 | −0.48 | 1.60 | 0.08 | 1.05 | 1,398 | 0.293 | 0 | 0 |
| | | | neglect | −0.26 | 0.44 | −1.13 | 0.62 | −0.04 | −0.58 | 1,398 | 0.564 | 0 | 0 |
| | | | interaction | −0.01 | 0.02 | −0.06 | 0.03 | −0.09 | −0.67 | 1,398 | 0.502 | 0 | 0 |
| LDS | GEN | SCR | abuse | 0.00 | 0.00 | 0.00 | 0.00 | −0.03 | −0.38 | 1,398 | 0.702 | 0 | 0 |
| | | | neglect | 0.00 | 0.00 | 0.00 | 0.00 | −0.06 | −0.77 | 1,398 | 0.443 | 0 | 0 |
| | | | interaction | 0.00 | 0.00 | 0.00 | 0.00 | 0.07 | 0.53 | 1,398 | 0.596 | 0 | 0 |
| | | Arousal ratings | abuse | −0.01 | 0.02 | −0.06 | 0.04 | −0.03 | −0.35 | 1,398 | 0.726 | 0 | 0 |
| | | | neglect | −0.01 | 0.02 | −0.05 | 0.03 | −0.04 | −0.50 | 1,398 | 0.618 | 0 | 0 |
| | | | interaction | 0.00 | 0.00 | 0.00 | 0.00 | 0.06 | 0.43 | 1,398 | 0.667 | 0 | 0 |
| | | Valence ratings | abuse | 0.00 | 0.02 | −0.05 | 0.05 | 0.00 | 0.02 | 1,398 | 0.982 | 0 | 0 |
| | | | neglect | 0.01 | 0.02 | −0.03 | 0.05 | 0.04 | 0.51 | 1,398 | 0.612 | 0 | 0 |
| | | | interaction | 0.00 | 0.00 | 0.00 | 0.00 | −0.04 | −0.34 | 1,398 | 0.738 | 0 | 0 |
| | | Contingency ratings | abuse | −0.04 | 0.30 | −0.63 | 0.55 | −0.01 | −0.14 | 1,398 | 0.891 | 0 | 0 |
| | | | neglect | −0.16 | 0.25 | −0.66 | 0.33 | −0.05 | −0.64 | 1,398 | 0.523 | 0 | 0 |
| | | | interaction | 0.01 | 0.01 | −0.02 | 0.03 | 0.07 | 0.52 | 1,398 | 0.603 | 0 | 0 |

*Appendix 1—table 7 Continued on next page*

*Appendix 1—table 7 Continued*

| Outcome | Phase | Measure | predictor | beta | SE_b | LL (95% CI) | UL (95% CI) | Beta | t | df | p | adj. R² | Cohen's f² |
|---|---|---|---|---|---|---|---|---|---|---|---|---|---|
| | | | abuse | 0.00 | 0.00 | 0.00 | 0.01 | 0.03 | 0.42 | 1,398 | 0.676 | 0 | 0 |
| | | SCR | neglect | 0.00 | 0.00 | −0.01 | 0.00 | −0.11 | −1.42 | 1,398 | 0.156 | 0 | 0 |
| | | | interaction | 0.00 | 0.00 | 0.00 | 0.00 | 0.04 | 0.28 | 1,398 | 0.778 | 0 | 0 |
| | | | abuse | 0.04 | 0.02 | 0.00 | 0.09 | 0.15 | 1.94 | 1,398 | 0.053 | 0 | 0 |
| | | Arousal ratings | neglect | 0.01 | 0.02 | −0.02 | 0.05 | 0.06 | 0.74 | 1,398 | 0.459 | 0 | 0 |
| | | | interaction | 0.00 | 0.00 | 0.00 | 0.00 | −0.19 | −1.50 | 1,398 | 0.133 | 0 | 0 |
| General reactivity | ALL | | abuse | 0.01 | 0.02 | −0.02 | 0.05 | 0.06 | 0.73 | 1,398 | 0.468 | 0 | 0 |
| | | Valence ratings | neglect | −0.01 | 0.01 | −0.04 | 0.02 | −0.05 | −0.69 | 1,398 | 0.493 | 0 | 0 |
| | | | interaction | 0.00 | 0.00 | 0.00 | 0.00 | −0.04 | −0.27 | 1,398 | 0.785 | 0 | 0 |
| | | | abuse | −0.10 | 0.26 | −0.60 | 0.40 | −0.03 | −0.40 | 1,180 | 0.689 | 0 | 0 |
| | | Contingency ratings | neglect | −0.17 | 0.22 | −0.60 | 0.25 | −0.07 | −0.80 | 1,180 | 0.424 | 0 | 0 |
| | | | interaction | 0.01 | 0.01 | −0.01 | 0.03 | 0.13 | 0.94 | 1,180 | 0.350 | 0 | 0 |

Note. ACQ = acquisition training, GEN = generalization phase, LDS = linear deviation score. Bold numbers indicate significant results (p<0.05).

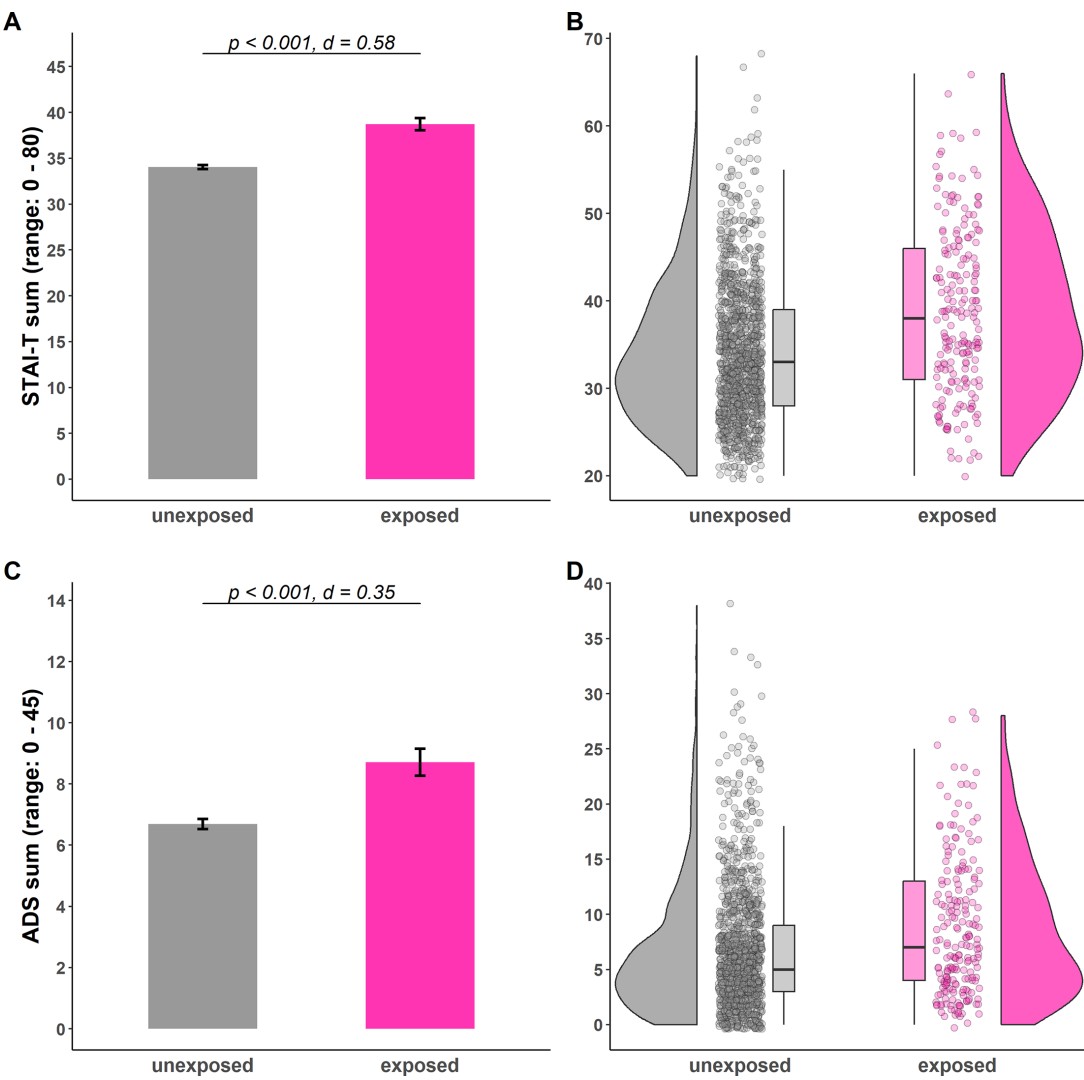

**Appendix 1—figure 6.** Illustration of STAI-T (**A–B**) and ADS-K (**C–D**) sum scores for individuals unexposed (gray) and exposed (pink) to childhood adversity. Barplots (**A and C**) with error bars represent means and standard errors of the means (SEMs) with $n_{unexposed}$ = 1199 and $n_{exposed}$ = 203, respectively. The statistical parameters presented in **A** and **C** are derived from Welch's tests. The a priori significance level was set to α = 0.05. Distributions of the data are illustrated in the raincloud plots (**B and D**). Points next to the densities represent the sum scores of each participant. Boxes of boxplots represent the interquartile range (IQR) crossed by the median as a bold line, ends of whiskers represent the minimum/maximum value in the data within the range of 25th/75th percentiles ± 1.5 IQR. STAI-T=Trait scale of the State-Trait Anxiety Inventory (*Spielberger, 1983*); ADS-K = Allgemeine Depressionsskala - Kurzform (short version of the Center for Epidemiological Studies-Depression Scale, CES-D; *Hautzinger and Bailer, 1993*).

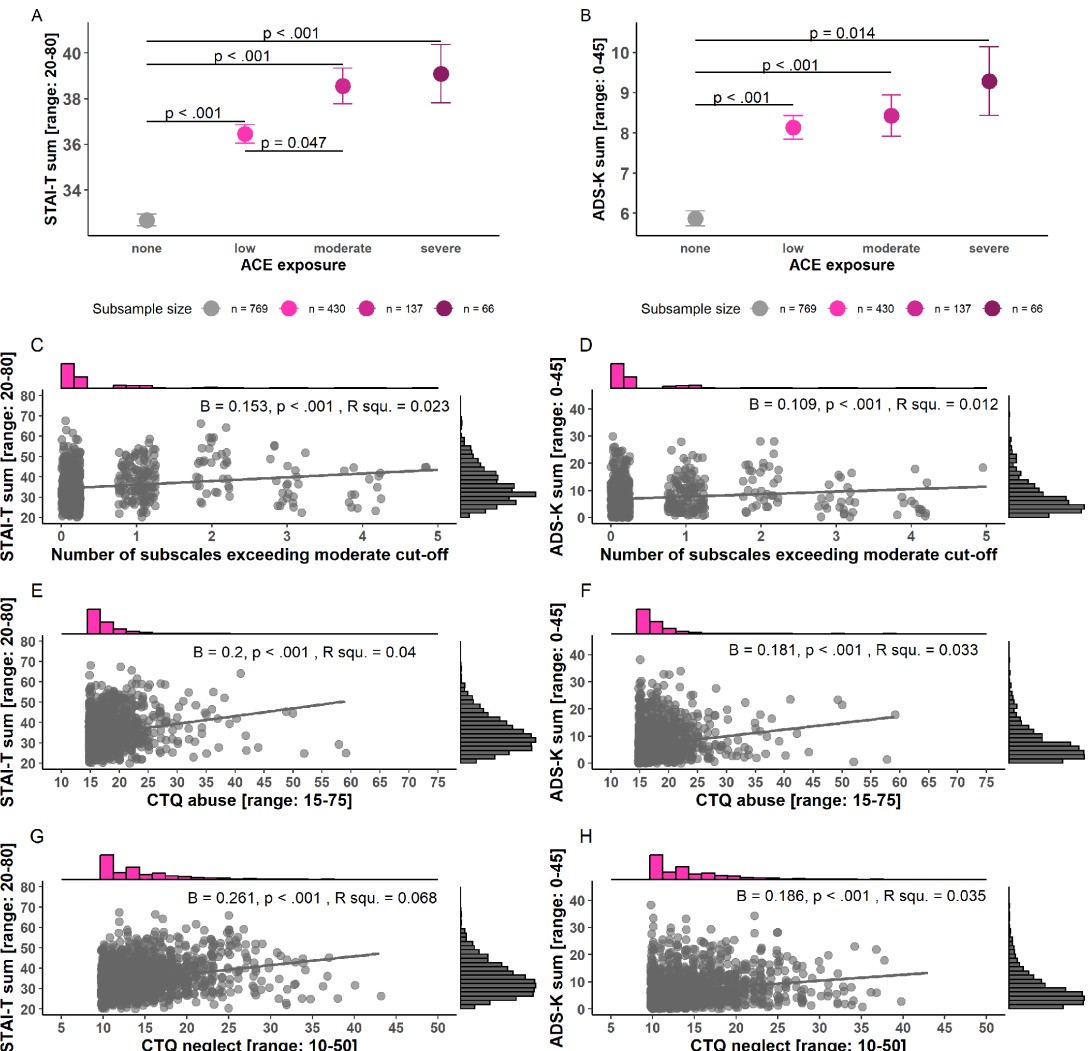

**Appendix 1—figure 7.** Illustration of associations of STAI-T (**A, C, E, G**) and ADS-K (**B, D, F, H**) with childhood adversity operationalized as cumulative risk models (**A-D**) and the specificity model (**E-H**). STAI-T=Trait scale of the State-Trait Anxiety Inventory (*Spielberger, 1983*); ADS-K = Allgemeine Depressionsskala - Kurzform (short version of the Center for Epidemiological Studies-Depression Scale, CES-D; *Hautzinger and Bailer, 1993*).

**Appendix 1—table 8.** Pearson correlations of STAI-T and ADS-K with skin conductance response (SCR).

|  | STAI-T | ADS-K |
|---|---|---|
| CS discrimination in SCR (ACQ) | *r*=–0.05, p=0.06 | *r*=**–0.057, p=0.033** |
| SCR to the CS+ (ACQ) | *r*=**–0.057, p=0.032** | *r*=**–0.057, p=0.032** |
| SCR to the CS- (ACQ) | *r*=–0.019, p=0.467 | *r*=–0.013, p=0.638 |
| CS discrimination in SCR (GEN) | *r*=0.001, p=0.964 | *r*=–0.032, p=0.234 |
| LDS in SCR | *r*=–0.045, p=0.091 | *r*=–0.038, p=0.153 |
| Mean reactivity in SCR | *r*=–0.019, p=0.484 | *r*=–0.006, p=0.835 |

Note. ACQ = acquisition training; GEN = generalization phase; STAI-T = Trait scale of the State-Trait Anxiety Inventory (**Spielberger, 1983**); ADS-K = Allgemeine Depressionsskala - Kurzform (short version of the Center for Epidemiological Studies-Depression Scale, [CES-D; **Hautzinger and Bailer, 1993**]); LDS = linear deviation score. Bold numbers indicate significant results (p<0.05).

