## [Editor Report · eLife Assessment]

This **important** study addresses two questions: (i) how danger signaling is altered for people with childhood adversities, and (ii) how this differs across different operationalizations of adversity. The latter is of particularly broad interest to multiple fields, given that childhood adversity is operationalized very differently across the literature. The study provides **compelling** evidence using a large sample size and rigorous statistical methods. These data will be of interest to scientists and clinicians interested in early life adversity, statistical approaches for quantifying stress exposure, or aversive learning.

---

## [Referee Report · Reviewer #1 (Public review)]

This is a very important paper, using a large dataset to definitively understand a phenomenon so far addressed using a range of diverging definitions and methods, typically with insufficient statistical power.

---

## [Referee Report · Reviewer #2 (Public review)]

Summary:

This important study uses convincing evidence to compare how different operationalizations of adverse childhood experience exposure related to patterns of skin conductance response during a fear conditioning task in a large sample of adults. Specifically, the authors compared the following operationalizations: dichotomization of the sample into "exposed" and "non-exposed" categories, cumulative adversity exposure, specificity of adversity exposure, and dimensional (threat versus deprivation) adversity exposure. The paper is thoughtfully framed and provides clear descriptions and rationale for procedures, as well as package version information and code. The authors' overall aim of translating theoretical models of adversity into statistical models, and comparing the explanatory power of each model, respectively, is an important and helpful addition to the literature.

Several outstanding strengths of this paper are the large sample size and its primary aim of statistically comparing leading theoretical models of adversity exposure in the context of skin conductance response. This paper also helpfully reports Cohen's d effect sizes, which aid in interpreting the magnitude of the findings. The methods and results are thorough and well-described.

---

## [Author Response]

**Joint Public Reviews:**
Here, the authors compare how different operationalizations of adverse childhood experience exposure related to patterns of skin conductance response during a fear conditioning task. They use a large dataset to definitively understand a phenomenon that, to date, has been addressed using a range of different definitions and methods, typically with insufficient statistical power. Specifically, the authors compared the following operationalizations: dichotomization of the sample into "exposed" and "non-exposed" categories, cumulative adversity exposure, specificity of adversity exposure, and dimensional (threat versus deprivation) adversity exposure. The paper is thoughtfully framed and provides clear descriptions and rationale for procedures, as well as package version information and code. The authors' overall aim of translating theoretical models of adversity into statistical models, and comparing the explanatory power of each model, respectively, is an important and helpful addition to the literature. However, the analysis would be strengthened by employing more sophisticated modelling techniques that account for between-subjects covariates and the presentation of the data needs to be streamlined to make it clearer for the broad audience for which it is intended.StrengthsSeveral outstanding strengths of this paper are the large sample size and its primary aim of statistically comparing leading theoretical models of adversity exposure in the context of skin conductance response. This paper also helpfully reports Cohen's d effect sizes, which aid in interpreting the magnitude of the findings. The methods and results are generally thorough.WeaknessesWeakness 1: The largest concern is that the paper primarily relies on ANOVAs and pairwise testing for its analyses and does not include between-subjects covariates. Employing mixedeffects models instead of ANOVAs would allow more sophisticated control over sources of random variance in the sample (especially important for samples from multi-site studies such as the present study), and further allow the inclusion of potentially relevant between-subjects covariates such as age (e.g. Eisenstein et al., 1990) and gender identity or sex assigned at birth (e.g. Kopacz II & Smith, 1971) (perhaps especially relevant due to possible to gender or sex-related differences in ACE exposure; e.g. Kendler et al., 2001). Also, proxies for socioeconomic status (e.g. income, education) can be linked with ACE exposure (e.g. Maholmes & King, 2012) and warrant consideration as covariates, especially if they differ across adversity-exposed and unexposed groups.

We appreciate the reviewer's suggestion and recognize the value of using (more) sophisticated statistical methods. However, we think that considerations which methods to employ should not only be guided by perceived complexity and think that the chosen ANOVA -based approach provides reliable and valid data. In our revision, we address the reviewer's suggestion by demonstrating that employing mixed models leaves the reported results unchanged (a). We would also like to refer the reviewer to the robustness analyses provided in the initial supplementary material (b).

a) Re-running analyses using mixed models

Based on the reviewers' suggestion, we repeated our main analyses (association between exposure to childhood adversity and SCRs, arousal, valence, and contingency ratings during fear acquisition and generalization) using linear mixed models, including age, sex, educational attainment, and childhood adversity as fixed effects, and site as a random effect. These analyses produced results similar to those in our manuscript, demonstrating a significant effect of childhood adversity on SCRs, as assessed by CS discrimination during both acquisition training and the generalization phase, and on general reactivity, but not on linear deviation scores (LDS). For the different rating types, we did not observe any significant effects of childhood adversity.

We would prefer to retain our main analyses as they are and report the linear mixed model results as additional results in the supplement. However, if the reviewer and editor have strong preferences otherwise, we are open to presenting the mixed models in the main manuscript and moving our previous analyses to the supplement.

We added the following paragraph to the main manuscript (page 25-26):

“At the request of a reviewer, we repeated our main analyses by using linear mixed models including age, sex, school degree (i.e. to approximate socioeconomic status), and exposure to childhood adversity as mixed effects as well as site as random effect. These analyses yielded comparable results demonstrating a significant effect of childhood adversity on CS discrimination during acquisition training and the generalization phase as well as on general reactivity, but not on the generalization gradients in SCRs (see Supplementary Table 2 A). Consistent with the results of the main analyses reported in our manuscript, we did not observe any significant effects of childhood adversity on the different types of ratings when using mixed models (see Supplementary Table 2 B-D). Some of the mixed model analyses showed significantly lower CS discrimination during acquisition training and generalization, and lower general reactivity in males compared to females (see Supplementary Table 2 for details).”

b) Additional robustness tests for the main analyses (already provided in the initial submission as supplementary material)

We would also like to refer the reviewer to the robustness analyses in the initial supplement to account for possible site effects. Adding site to the analyses affected the pvalue in only one instance: entering site as covariate in analyses of CS discrimination during acquisition training attenuated the p-value of the ACQ exposure effect from p = 0.020 to p = 0.089.

Further robustness checks involved repeating our main analyses while excluding (a) physiological non-responders (participants with only SCRs = 0) and (b) extreme outliers (data points ± 3 SDs from the mean) to ensure generalizable results. These repetitions of the analyses did not lead to any changes in the results.

We did not include age in our primary analyses due to the homogeneity of our sample and the lack of related hypotheses. Additionally, socio-economic status was assessed only crudely via the highest education level attained, rendering it of limited use.

Weakness 2: On a related methodological note, the authors mention that scores representing threat and deprivation were not problematically collinear due to VIFs being <10; however, some sources indicate that VIFs should be <5 (e.g. Akinwande et al., 2015).

We thank the reviewer for bringing different cut-offs to our attention. We have revised this section to highlight the arbitrary nature of their interpretation (page 33):

“Within the dimensional model framework, the issue of multicollinearity among predictors (i.e., different childhood adversity types) is frequently discussed (McLaughlin et al., 2021; Smith & Pollak, 2021). If we apply the rule of thumb of a variance inflation factor (VIF) > 10, which is often used in the literature to indicate concerning multicollinearity (e.g., Hair, Anderson, Tatham, & Black, 1995; Mason, Gunst, & Hess, 1989; Neter, Wasserman, & Kutner, 1989), we can assume that that multicollinearity was not a concern in our study (abuse: VIF = 8.64; neglect: VIF = 7.93). However, some authors state that VIFs should not exceed a value of 5 (e.g., Akinwande, Dikko, and Samson (2015)), while others suggest that these rules of thumb are rather arbitrary (O’brien, 2007).”

Weakness 3: Additionally, the paper reports that higher trait anxiety and depression symptoms were observed in individuals exposed to ACEs, but it would be helpful to report whether patterns of SCR were in turn associated with these symptom measures and whether the different operationalizations of ACE exposure displayed differential associations with symptoms.

We thank the reviewer for highlighting these relevant points. We have included additional analyses in the supplementary material in response to this comment. Figures and the corresponding text are also copied below for your convenience.

We added the following paragraphs to the main manuscript: Methods (page 21):

“Analyses of trait anxiety and depression symptoms

To further characterize our sample, we compared individuals being unexposed compared to exposed to childhood adversity on trait anxiety and depression scores by using Welch tests due to unequal variances.

On the request of a reviewer, we additionally investigated the association of childhood adversity as operationalized by the different models used in our explanatory analyses (i.e., cumulative risk, specificity, and dimensional model) and trait anxiety as well as depression scores (see Supplementary Figure 7). By using STAI-T and ADS-K scores as independent variable, we calculated (a) a comparison of conditioned responding of the four severity groups (i.e., no, low, moderate, severe exposure to childhood adversity) using one-way ANVOAs and the association with the number of sub-scales exceeding an at least moderate cut-off in simple linear regression models for the implementation of the cumulative risk model, and (b) the association with the CTQ abuse and neglect composite scores in separate linear regression models for the implementation of the specificity/dimensional models. On request of the reviewer, we also calculated the Pearson correlation between trait anxiety (i.e., STAI-T scores), depression scores (i.e., ADS-K scores) and conditioned responding in SCRs (see Supplementary Table 8).”

Results (page 38):

“Analyses of trait anxiety and depression symptoms

As expected, participants exposed to childhood adversity reported significantly higher trait anxiety and depression levels than unexposed participants (all *p*’s < 0.001; see Table 1 and Supplementary Figure 6). This pattern remained unchanged when childhood adversity was operationalized differently - following the cumulative risk approach, the specificity, and dimensional model (see methods). These additional analyses all indicated a significant positive relationship between exposure to childhood adversity and trait anxiety as well as depression scores irrespective of the specific operationalization of “exposure” (see Supplementary Figure 7).

CS discrimination during acquisition training and the generalization phase, generalization gradients, and general reactivity in SCRs were unrelated to trait anxiety and depression scores in this sample with the exception of a significant association between depression scores and CS discrimination during fear acquisition training (see Supplementary Table 8). More precisely, a very small but significant negative correlation was observed indicating that high levels of depression were associated with reduced levels of CS discrimination (r = -0.057, p = 0.033). The correlation between trait anxiety levels and CS discrimination during fear acquisition training was not statistically significant but on a descriptive level, high anxiety scores were also linked to lower CS discrimination scores (r = -0.05, p = 0.06) although we highlight that this should not be overinterpreted in light of the large sample. However, both correlations (i.e., CS-discrimination during fear acquisition training and trait anxiety as well as depression, respectively) did not statistically differ from each other (z = 0.303, p = 0.762, Dunn & Clark, 1969). Interestingly, and consistent with our results showing that the relationship between childhood adversity and CS discrimination was mainly driven by significantly lower CS+ responses in exposed individuals, trait anxiety and depression scores were significantly associated with SCRs to the CS+, but not to the CS- during acquisition training (see Supplementary Table 8).”

Weakness 4: Given the paper's framing of SCR as a potential mechanistic link between adversity and mental health problems, reporting these associations would be a helpful addition. These results could also have implications for the resilience interpretation in the discussion (lines 481-485), which is a particularly important and interesting interpretation.

We have added a paragraph on this to the discussion (page 41):

“Interestingly, in our study, trait anxiety and depression scores were mostly unrelated to SCRs, defined by CS discrimination and generalization gradients based on SCRs as well as general SCR reactivity, with the exception of a significant - albeit minute - relationship between CS discrimination during acquisition training and depression scores (see above). Although reported associations in the literature are heterogeneous (Lonsdorf et al., 2017), we may speculate that they may be mediated by childhood adversity. We conducted additional mediation analyses (data not shown) which, however, did not support this hypothesis. As the potential links between reduced CS discrimination in individuals exposed to childhood adversity and the developmental trajectories of psychopathological symptoms are still not fully understood, future work should investigate these further in - ideally - prospective studies.”

Weakness 5: Given that the manuscript criticizes the different operationalizations of childhood adversity, there should be greater justification of the rationale for choosing the model for the main analyses. Why not the 'cumulative risk' or 'specificity' model? Related to this, there should also be a stronger justification for selecting the 'moderate' approach for the main analysis. Why choose to cut off at moderate? Why not severe, or low? Related to this, why did they choose to cut off at all? Surely one could address this with the continuous variable, as they criticize cut-offs in Table 2.

We thank the reviewers and editors for bringing to our attention that our reasoning for choosing the main model was not clear. As outlined in the manuscript, we chose the approach for the main analyses from the literature as a recent review on this topic (Ruge et al., 2023) has shown the moderate CTQ cut-off to be the most abundantly employed in the field of research on associations between childhood adversity and threat learning. We have made this rationale more explicit in our revised manuscript (page 15/21):

“Operationalization of "exposure"

We implemented different approaches to operationalize exposure to childhood adversity in the main analyses and exploratory analyses (see Table 2). In the main analyses, we followed the approach most commonly employed in the field of research on childhood adversity and threat learning - using the moderate exposure cut-off of the CTQ (for a recent review see Ruge et al. (2024)). In addition, the heterogeneous operationalizations of classifying individuals into exposed and unexposed to childhood adversity in the literature (Koppold, Kastrinogiannis, Kuhn, & Lonsdorf, 2023; Ruge et al., 2024) hampers comparison across studies and hence cumulative knowledge generation. Therefore, we also provide exploratory analyses (see below) in which we employ different operationalizations of childhood adversity exposure.”

“Exploratory analyses

Additionally, the different ways of classifying individuals as exposed or unexposed to childhood adversity in the literature (Koppold et al., 2023; for discussion see Ruge et al., 2024) hinder comparison across studies and hence cumulative knowledge generation. Therefore, we also conducted exploratory analyses using different approaches to operationalize exposure to childhood adversity (see Table 2 for details).”

Furthermore, as correctly noted, we fully agree that employing the moderate cut-off (or any cut-off in fact) is in principle an arbitrary decision - despite being guided by and derived from the literature in the field. However, we would like to draw the reviewers’ attention to Figure 5 in the initial submission (please see also below): Although the differences in SCR between severity groups were not significant, the overall pattern suggests at a descriptive level that the decline in CS discrimination, LDS and general reactivity in SCR occurs mainly when childhood adversity exceeds a moderate level. Thus, while we used the moderate cut-off as it was recently shown to be the most widely used approach in the literature (see Ruge et al., 2023), our exploratory analyses also seem to suggest on a descriptive level, that this cut-off may indeed “make sense”. We also refer to this in the results section (page 31-32) and discussion (page 43-44):

Results:

“However, on a descriptive level (see Figure 5), it seems that indeed exposure to at least a moderate cut-off level may induce behavioral and physiological changes (see main analysis, Bernstein & Fink, 1998). This might suggest that the cut-off for exposure commonly applied in the literature (see Ruge et al., 2024) may indeed represent a reasonable approach.”

Discussion:

“It is noteworthy, however, that this cut-off appears to map rather well onto psychophysiological response patterns observed here (see Figure 5). More precisely, our exploratory results of applying different exposure cut-offs (low, moderate, severe, no exposure) seem to indicate that indeed a moderate exposure level is “required” for the manifestation of physiological differences, suggesting that childhood adversity exposure may not have a linear or cumulative effect.”

Weakness 6: In the Introduction, the authors predict less discrimination between signals of danger (CS+) and safety (CS-) in trauma-exposed individuals driven by reduced responses to the CS+. Given the potential impact of their findings for a larger audience, it is important to give greater theoretical context as to why CS discrimination is relevant here, and especially what a reduction in response specifically to danger cues would mean (e.g. in comparison to anxiety, where safety learning is impacted).

We thank the reviewer for highlighting that this was not sufficiently clear. We revised the paragraph in the introduction as follows (page 7-8):

“Fear acquisition as well as extinction are considered as experimental models of the development and exposure-based treatment of anxiety- and stress-related disorders. Fear generalization is in principle adaptive in ensuring survival (“better safe than sorry”), but broad overgeneralization can become burdensome for patients. Accordingly, maintaining the ability to distinguish between signals of danger (i.e., CS+) and safety (i.e., CS-) under aversive circumstances is crucial, as it is assumed to be beneficial for healthy functioning (Hölzel et al., 2016) and predicts resilience to life stress (Craske et al., 2012), while reduced discrimination between the CS+ and CS- has been linked to pathological anxiety (Duits et al., 2015; Lissek et al., 2005): Meta-analyses suggest that patients suffering from anxiety- and stress-related disorders show enhanced responding to the safe CS- during fear acquisition (Duits et al., 2015). During extinction, patients exhibit stronger defensive responses to the CS+ and a trend toward increased discrimination between the CS+ and CS- compared to controls, which may indicate delayed and/or reduced extinction (Duits et al., 2015). Furthermore, meta-analytic evidence also suggests stronger generalization to cues similar to the CS+ in patients and more linear generalization gradients (Cooper, van Dis, et al., 2022; Dymond, Dunsmoor, Vervliet, Roche, & Hermans, 2015; Fraunfelter, Gerdes, & Alpers, 2022). Hence, aberrant fear acquisition, extinction, and generalization processes may provide clear and potentially modifiable targets for intervention and prevention programs for stress-related psychopathology (McLaughlin & Sheridan, 2016).”

**Recommendations for the authors:**
Abstract:Comment 1:(a) It does not succinctly describe the background rationale well (i.e. it tries to say too much). It should be streamlined. There is a lot of 'jargon', which muddies the results, and too many concepts are introduced at each part and assume knowledge from the reader.

We thank the reviewer for providing constructive guidance for revisions. We have revised our abstract according to these suggestions.

(b) Multiple terms for childhood trauma are used: ACEs, early adversity, childhood trauma, and childhood maltreatment. Choose one term and stick to it to enhance clarity. Why not just use childhood adversity, as in the title? Related to this, the use of ACEs sets up an expectation that ACE questionnaire was used, so readers are then surprised to find they used the childhood trauma questionnaire.

We thank the reviewer for bringing this to our attention. As suggested by the reviewer, we use the term “childhood adversity” in our revised manuscript.

Introduction:Comment 2:The phrasing seems to 'exaggerate' the trauma problem and is too broad in the first paragraph - e.g., "two-thirds of people experience one or more traumatic events..." It is important to clarify that not all of these people will go on to develop behavioral, somatic, and psychopathological conditions. Could break this down more into how many people have low, moderate, or severe for clarity, as 1 childhood adversity is different to 5+, and the type.

We thank the reviewer for bringing this to our attention and have revised the first paragraph accordingly (page 6). Please note, however, that in the literature typically a specific cut-off (e.g. moderate) is used and the number of individuals that would meet different cut-offs (e.g., low and high) are not specifically reported.

“Exposure to childhood adversity is rather common, with nearly two thirds of individuals experiencing one or more traumatic events prior to their 18th birthday (McLaughlin et al., 2013). While not all trauma-exposed individuals develop psychopathological conditions, there is some evidence of a dose-response relationship (Danese et al., 2009; Smith & Pollak, 2021; Young et al., 2019). As this potential relationship is not yet fully clear, understanding the mechanisms by which childhood adversity becomes biologically embedded and contributes to the pathogenesis of stress-related somatic and mental disorders is central to the development of targeted intervention and prevention programmes.”

Comment 3:The published cut-offs for exposed/unexposed should be indicated here.

We have included the published cut-offs as suggested (page 10):

We operationalize childhood adversity exposure through different approaches: Our main analyses employ the approach adopted by most publications in the field (see Ruge et al., 2024 for a review) - dichotomization of the sample into exposed vs. unexposed based on published cut-offs for the Childhood Trauma Questionnaire [CTQ; Bernstein et al. (2003); Wingenfeld et al. (2010)]. Individuals were classified as exposed to childhood adversity if at least one CTQ subscale met the published cut-off (Bernstein & Fink, 1998; Häuser, Schmutzer, & Glaesmer, 2011) for at least moderate exposure (i.e., emotional abuse 13, physical abuse 10, sexual abuse 8, emotional neglect 15, physical neglect 10).

Comment 4:Please check for overly complex sentences, and reduce the complexity. For example: "In addition, we provide exploratory analyses that attempt to translate dominant (verbal) theoretical accounts (McLaughlin et al., 2021; Pollak & Smith, 2021) on the impact of exposure to ACEs into statistical tests while acknowledging that such a translation is not unambiguous and these exploratory analyses should be considered as showcasing a set of plausible solutions."

We have revised this section and carefully proofread our manuscript by paying attention to this (page 10):

“In addition, we provide exploratory analyses that attempt to translate dominant (verbal) theoretical accounts (McLaughlin et al., 2021; Pollak & Smith, 2021) on the impact of exposure to childhood adversity into statistical tests. At the same time, we acknowledge that such a translation is not unambiguous and these exploratory analyses should be considered as showcasing a set of plausible solutions”

Here is another example of reducing the complexity of our sentences (page 6):

“Learning is a core mechanism through which environmental inputs shape emotional and cognitive processes and ultimately behavior. Thus, learning mechanisms are key candidates potentially underlying the biological embedding of exposure to childhood adversity and their impact on development and risk for psychopathology (McLaughlin & Sheridan, 2016).”

Methods:Comment 5:Is this study part of a larger project? These outcomes were probably not the primary outcomes of this multicenter project. The readers need to understand how this (crosssectional?) analysis was nested in this larger trial.

We thank the reviewers and editor for bringing to our attention that this was not sufficiently clear. Thus far, we included the information that we used the participants recruited for large multicentric study in the main manuscript, but point to the inclusion of more information in the supplement (page 11):

“In total, 1678 healthy participants (age_M_ = 25.26 years, age_SD_ = 5.58 years, female = 60.10%, male = 39.30%) were recruited in a multi-centric study at the Universities of Münster, Würzburg, and Hamburg, Germany (SFB TRR58). Data from parts of the Würzburg sample have been reported previously (Herzog et al., 2021; Imholze et al., 2023; Schiele, Reinhard, et al., 2016; Schiele, Ziegler, et al., 2016; Stegmann et al., 2019). These previous reports, also those focusing on experimental fear conditioning (Schiele, Reinhard, et al., 2016; Stegmann et al., 2019), addressed, however, research questions different from the ones investigated here (see also Supplementary Material for details).”

Moreover, we have included additional information on the larger trial in our revised supplement (page 2):

“Participants of this study were recruited in a multi-centric collaborative research center “Fear, anxiety, anxiety disorders” joining forces between the Universities of Hamburg,

Würzburg, and Münster, Germany (SFB TRR58). During the second funding period of (20132016), all three sites recruited a large sample (N ~500) in the context of the Z project. All participants underwent the cross-sectional experimental paradigm reported here and were additionally extensively characterized to allow specific subprojects to recruit target subpopulations serving different aims with a focus on molecular genetic, epigenetic, or other research questions (see Herzog et al. (2021); Imholze et al. (2023); Schiele, Reinhard, et al. (2016); Schiele, Ziegler, et al. (2016); Stegmann et al. (2019)). The question on the association of exposure to childhood adversity and recent adversity was part of the primary research question of one subproject led by the senior author of this work (B07, TBL) and was hence a research question of primary interest also for this multicentric project.”

Comment 6:Table 1 does not include percentages (a reader must calculate them: for example, 15% exposed?). These numbers belong in the results (i.e., it is confusing to read about the exposed/non-exposed before we know how it has been calculated).

We have added the percentages as suggested and have included information on how exposed and unexposed was calculated as a table caption. We have considered moving the table to the results section but find it more suitable here.

Comment 7:A procedure figure could be useful.

We thank the reviewer for this advice and have included a procedure figure in the supplementary material.

Comment 8:Physiological data recordings and processing paragraph: The reasoning as to why the authors chose log transformation over square root transformation, or an approach that does not require transformation is not clear.

We thank the reviewer for notifying us that we did not make this point clear enough. We opted for a log-transformation and range-correction of the SCR data because we use these transformations consistently in our laboratory (e.g., Ehlers et al., 2020; Kuhn et al., 2016; Scharfenort & Lonsdorf, 2016; Sjouwerman et al., 2015; Sjouwerman et al. 2020). In addition, log-transformed and range-corrected data are assumed to be closer to a normal distribution, to have a lower error variance resulting in larger effect sizes (Lykken & Venables, 1971; Lykken, 1972; Sjouwerman et al., 2022), and appear to have - at least descriptively - higher reliability compared to raw data (Klingelhöfer-Jens et al., 2022). We added a sentence on this to the methods section (page 14):

Note that previous work using this sample (Schiele, Reinhard, et al., 2016; Stegmann et al., 2019) had used square-root transformations but we decided to employ a log-transformation and range-correction (i.e., dividing each SCR by the maximum SCR per participant). We used log-transformation and range-correction for SCR data because these transformations are standard practice in our laboratory and we strive for methodological consistency across different projects (e.g., Ehlers, Nold, Kuhn, Klingelhöfer-Jens, & Lonsdorf, 2020; Kuhn, Mertens, & Lonsdorf, 2016; Scharfenort, Menz, & Lonsdorf, 2016; Sjouwerman & Lonsdorf, 2020; Sjouwerman, Niehaus, & Lonsdorf, 2015). Additionally, log-transformed and rangecorrected data are generally assumed to approximate a normal distribution more closely and exhibit lower error variance, which leads to larger effect sizes (Lykken, 1972; Lykken & Venables, 1971; Sjouwerman, Illius, Kuhn, & Lonsdorf, 2022). Additionally, on a descriptive level, this combination of transformations appear to offer greater reliability compared to using raw data alone (Klingelhöfer-Jens, Ehlers, Kuhn, Keyaniyan, & Lonsdorf, 2022).

Ehlers, M. R., Nold, J., Kuhn, M., Klingelhöfer-Jens, M., & Lonsdorf, T. B. (2020). Revisiting potential associations between brain morphology, fear acquisition and extinction through new data and a literature review. *Scientific Reports*, *10*(1), 19894. https://doi.org/10.1038/s41598-020-76683-1

Kuhn, M., Mertens, G., & Lonsdorf, T. B. (2016). State anxiety modulates the return of fear. *International Journal of Psychophysiology: Official Journal of the International Organization of Psychophysiology*, *110*, 194–199. https://doi.org/10.1016/j.ijpsycho.2016.08.001

Scharfenort, R., & Lonsdorf, T. B. (2016). Neural correlates of and processes underlying generalized and differential return of fear. *Social Cognitive and Affective Neuroscience*, *11*(4), 612–620. https://doi.org/10.1093/scan/nsv142

Sjouwerman, R., Niehaus, J., & Lonsdorf, T. B. (2015). Contextual Change After Fear Acquisition Affects Conditioned Responding and the Time Course of Extinction Learning—Implications for Renewal Research. *Frontiers in Behavioral Neuroscience*, *9*. https://doi.org/10.3389/fnbeh.2015.00337

Sjouwerman, R., Scharfenort, R., & Lonsdorf, T. B. (2020). Individual differences in fear acquisition: Multivariate analyses of different emotional negativity scales, physiological responding, subjective measures, and neural activation. *Scientific Reports*, *10*(1), 15283. https://doi.org/10.1038/s41598-020-72007-5

Comment 9:There are 24 lines of text of R packages. I do not think this is necessary for the manuscript document and could be moved to the Supplement.

We thank the reviewer for this comment and understand that it may take a considerable amount of space to list all the references of the R packages. However, we think it is important to prominently credit the respective authors of the R packages. Yet, if this is an important concern of the reviewer and editor, we will reconsider this point.

Comment 10:It is not clear why the authors chose to analyze summary scores across trials rather than including a time factor for the acquisition phase.

We would like to thank the reviewer for highlighting that the factor time may be interesting as well. However, we think that in our case the time factor is less interesting, as the acquisition effect itself is rather strong. Nevertheless, we have included a figure in the supplement that shows the time course of the SCR by displaying trial-by-trial data across the acquisition and generalization phase for transparency. This figure (Supplementary figure 4) shows that the trajectories appear to barely differ between individuals who were unexposed vs. exposed to moderate childhood adversity. Hence, we think that the analysis approach we have chosen is unlikely to overshadow central time-depending effects. However, if the reviewer and editor has strong feelings about this point, we will consider integrating additional analyses including the time factor in the supplement.

Results:Comment 11:The caption of Figure 3 does not match the figure. Please check this.

We thank the reviewers and editor for attentive reading and have revised this part.

References:Comment 12:The Ruge et al paper that is cited many times throughout does not have a valid DOI in the References section. Additionally, the author list on the preprint server is substantially different from that listed in the manuscript. Please correct this reference.

We thank the reviewers and editor for attentive reading and have corrected this reference. The provided doi was functioning at our end and we hope that this now also applies to the reviewers.